# pLSTM: parallelizable Linear Source Transition Mark networks

**Korbinian Pöppel**[1,2]     **Richard Freinschlag**[1]     **Thomas Schmied**[1]     **Wei Lin**[1]

**Sepp Hochreiter**[1,3]

[1] ELLIS Unit Linz, LIT AI Lab, Institute for Machine Learning
Johannes Kepler University Linz, Austria
[2] Zuse School ELIZA
[3] NXAI GmbH
poeppel@ml.jku.at

## Abstract

Modern recurrent architectures, such as xLSTM and Mamba, have recently challenged the Transformer in language modeling. However, their structure constrains their applicability to sequences only or requires processing multi-dimensional data structures, such as images or molecular graphs, in a pre-defined sequential order. In contrast, Multi-Dimensional RNNs (MDRNNs) are well suited for data with a higher level structure, like 2D grids, trees, and directed acyclic graphs (DAGs). In this work, we extend the notion of multi-dimensionality to linear RNNs. We introduce parallelizable Linear Source Transition Mark networks (pLSTMs) using Source, Transition, and Mark gates that act on the linegraph of a general DAG. This enables parallelization in analogy to parallel associative scans and the chunkwise-recurrent form of sequential linear RNNs, but for DAGs. For regular grids (1D and 2D), like images, this scheme can be efficiently implemented using einsum operations, concatenations, and padding in logarithmic time. pLSTMs tackle the vanishing/exploding activation/gradient problem for long distances in DAGs via two distinct modes: a directed propagation mode (P-mode) and a diffusive distribution mode (D-mode). To showcase the long-range capabilities of pLSTM, we introduce arrow-pointing extrapolation as a synthetic computer vision task that contains long-distance directional information. We demonstrate that pLSTMs generalize well to larger image sizes, whereas Transformers struggle to extrapolate. On established molecular graph and computer vision benchmarks, pLSTMs also show strong performance. The complete code is available at
https://github.com/ml-jku/plstm_experiments.

## 1 Introduction

Linear RNNs such as DeltaNet [Schlag et al., 2021], Gated Linear Attention [Yang et al., 2023], Mamba [Gu and Dao, 2023, Dao and Gu, 2024], and xLSTM (mLSTM) [Beck et al., 2025b] have recently evolved as a powerful alternative to the Transformer [Vaswani et al., 2017]. In addition to sequence-parallelizable training, such modern recurrent architectures are more efficient at inference time than Transformers. In contrast to classical RNNs, they come with associative memory expansion, enabling more performant sequence modeling. However, these modern recurrent architectures are inherently limited to sequences. While linear RNNs have shown good performance for multi-dimensional data such as images, they enforce a pre-defined sequential traversal structure [Zhu

39th Conference on Neural Information Processing Systems (NeurIPS 2025).

et al., 2024, Alkin et al., 2025]. To illustrate, for images, this corresponds to processing patches in a scanline form. However, this is problematic as, within one layer, even (vertical) neighbors in a patch grid are only distantly related, in the (horizontal line-wise) processing. Even when switching between a line- and column-wise order across layers, diagonal relationships are not covered sufficiently. This mismatch of short-range spatial and long-distance relations in a certain path with precise positioning requirements leads to credit assignment problems [Minsky, 1961, Bengio and Frasconi, 1993, Schmidhuber, 2015] and suffers from vanishing activations/gradients [Hochreiter, 1991, Bengio et al., 1994, Hochreiter et al., 2001, Hochreiter and Schmidhuber, 1997, Pascanu et al., 2013].

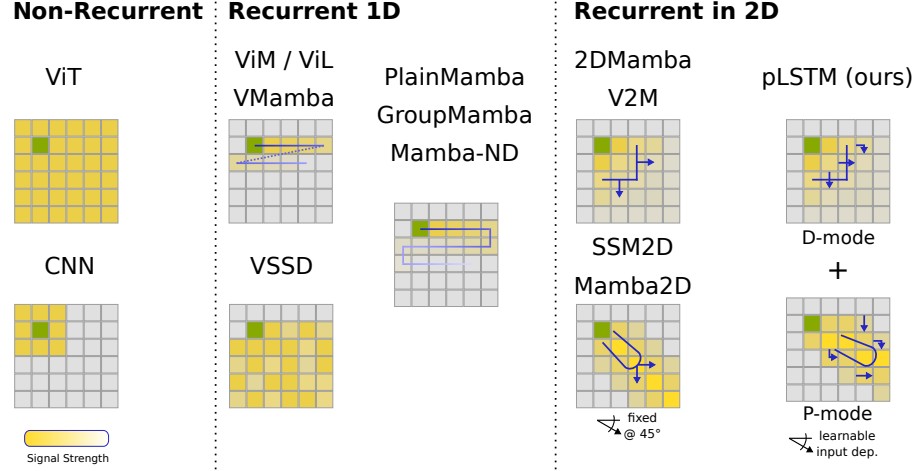

**Figure 1:** Illustration of the **receptive fields** induced by pLSTM and related architectures (for a single layer). CNNs are locally restricted while ViTs have a global receptive field. Modern recurrent architectures, such as ViM, traverse the 2D grid sequentially. pLSTM effectively extends the receptive field via its combination of D-mode and P-mode.

Multi-dimensional RNNs (MDRNNs) [Graves et al., 2007] have demonstrated how non-linear RNNs like LSTMs [Hochreiter, 1991, Hochreiter and Schmidhuber, 1997] can be extended to multi-dimensional data, such as images [Graves et al., 2007, van den Oord et al., 2016], trees [Tai et al., 2015], and DAGs [Peng et al., 2017]. In this work, we translate the findings of MDRNNs to linear RNNs and introduce parallelizable Linear Source Transition Mark Networks (pLSTM). To achieve this, we first translate their input, forget, and output gates to Source, Transition, and Mark gates, respectively. This is to make the resulting structure applicable to general DAGs. Second, we derive a parallelization scheme enabled by the previous adaptation, leading to higher-order Source, Transition, and Mark terms - forming a parallel associative scan in multiple dimensions and DAGs. This also allows for a chunkwise-parallel formulation known from linear RNNs [Yang et al., 2023, Beck et al., 2025a], using the parallelization up to a certain level and the recurrent option between chunks/patches. Third, we derive two stabilization options, the P-mode for directional propagation and the D-mode for undirected global distribution. In Figure 1, we illustrate how the two modes affect the receptive field of pLSTM compared to other linear RNNs. We use pLSTM with MLP layers in a residual pre-LayerNorm backbone, resembling a Transformer with replaced multi-head attention.

To showcase the long-range capabilities of pLSTM in multiple dimensions, we first introduce the synthetic arrow-pointing extrapolation task (see Figure 2). As the directional information is not tied to only testing horizontal and vertical cases separately, a certain line-wise or column-wise ordering (even in alternation) cannot capture the directional information in the sequential RNN transitions, at least for a limited number of layers. Importantly, pLSTM generalizes well to increasing image resolutions compared to the Transformer (see Section 5.1). Moreover, we demonstrate the efficacy of pLSTM on established computer-vision and graph benchmarks. Experiments on ImageNet-1k (see Section 5.2) and on molecular graphs (see Section 5.4) show promising results compared to baselines and good scaling behavior to larger model sizes.

In summary, we make the following **contributions**:

- We translate the findings of classic MDRNNs to linear RNNs and introduce pLSTM, which comes with adapted gates and a scalable chunkwise-parallel formulation.
- We formally derive the general stabilization of pLSTMs for long-range propagation on general DAGs (including images), establishing the P- and D-mode cases.
- We introduce the synthetic Arrow Pointing Task to highlight the theoretical advantage of pLSTMs, in which pLSTM shows strong extrapolation abilities, and provide a highly scalable implementation of pLSTM.

## 2 Related work

**Modern Recurrent Architectures**  This work presents a new form of linear RNNs, where DeltaNet [Schlag et al., 2021, Yang et al., 2024], LRU [Orvieto et al., 2023], GLA [Yang et al., 2023], Mamba [Gu and Dao, 2023, Dao and Gu, 2024], and xLSTM [Beck et al., 2025b] (in the mLSTM form) have shown their effectiveness on sequence modeling in the language domain. Recently, this line of work has been complemented by TTT [Sun et al., 2024], Titans [Behrouz et al., 2024], and DeltaProduct [Siems et al., 2025], which motivate this structure as a gradient-based optimization in context, in line with early work on Fast-Weight Programmers [Schlag et al., 2021]. Due to their ability for parallelization during training, modern recurrent architectures scale to large-scale datasets similar to Transformers. Moreover, they come with efficient inference, which is attractive for real-world applications [Schmidinger et al., 2024, Schiff et al., 2024, Schmied et al., 2024].

**Non-Linear Multi-Dimensional RNNs**  The first foundational extension of non-linear RNNs / LSTMs to multiple dimensions was carried out by Graves et al. [2007]. Subsequently, Stacked LSTM was proposed [Graves et al., 2013], which stacks LSTM layers on top of each other. MDRNNs are hard to parallelize, while re-arranging the traditional cuboid order of computations in MDLSTMs in pyramidal fashion led to PyraMiD LSTM [Stollenga et al., 2015], which can be parallelized. Grid LSTM [Kalchbrenner et al., 2015] operates in multiple processing dimensions simultaneously, thus generalizing one-dimensional LSTM to more dimensions. Tree-LSTM [Tai et al., 2015] and DAG-LSTM [Zhu et al., 2016, Peng et al., 2017, Chen et al., 2017] extend MDRNNs to tree structures and DAGs. PixelRNN [Van Den Oord et al., 2016] operates directly on pixel-level and improves MDRNNs for images. Due to their strictly recurrent nature, these architectures are not parallelizable and therefore unsuitable for modern hardware. Moreover, they lack the associative memory component (state expansion) of modern linear RNNs, making them less powerful.

**Linear RNNs on Multi-Dimensional data**  VisionMamba (ViM) [Zhu et al., 2024] and Vision-LSTM (ViL) [Alkin et al., 2025] applied linear RNNs to the vision domain to challenge the common Vision Transformer (ViT) with its quadratic scaling [Dosovitskiy et al., 2021]. These works rely on traversing the 2D plane in a predefined order to accommodate their sequential nature and often employ flipping the traversal order across layers. This was also investigated further in Mamba-ND [Li et al., 2025]. Similarly, several works use Mamba on graph data, covering the graph with multiple paths [Wang et al., 2024a] or forming a sequence in other ways [Behrouz and Hashemi, 2024], with an overview presented in Atitallah et al. [2024]. To

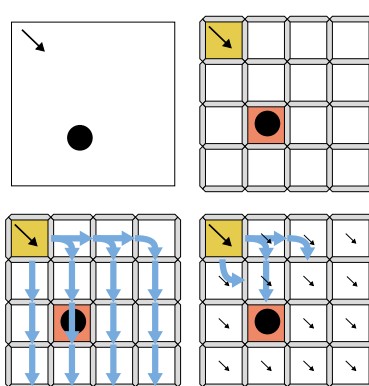

**Figure 2:** Illustration of the **Arrow pointing task**. The model has to classify whether an arrow is pointing towards a circle (top left). Models with global receptive fields, such as Vision Transformers (ViTs), can solve this task by leveraging directional information encoded via positional embeddings (top right), but they often struggle to generalize to higher resolutions. In contrast, pLSTMs can effectively solve this task in both diffusive distribution (D-mode, bottom left) and directed propagation (P-mode, bottom right) modes, enabling long-range reasoning and better scalability.

the best of our knowledge, all of these rely on a sequential representation to apply Mamba at some point, and therefore do not use the graph's structural information for the recurrence. Recent works extend the Mamba architecture to two dimensions. Most related to our work are V2M [Wang et al.,

2024b] and 2DMamba [Zhang et al., 2024a], which cover only the D-mode of pLSTM, at the loss of directional propagation. Mamba2D [Baty et al., 2024] and 2-D SSM [Baron et al., 2024] cover the P-mode of pLSTM, but in contrast to pLSTM, they are limited to the diagonal line instead of input-directed propagation along any line because of their fixed normalization factor of $1/2$ (see Appendix D.2).

**Relation to GNNs and MPNNs -** Since pLSTM propagates recurrent states along a (directed acyclic) graph, it can be seen as a form of Graph Neural Network (GNN) [Scarselli et al., 2009] that covers a whole connected component of a DAG per layer instead of just a one-hop neighborhood.

## 3 Background

### 3.1 Multidimensional RNNs

The most general form of MDRNNs[Graves et al., 2007] was introduced as DAG LSTM by Zhu et al. [2016], as it can be applied to all directed acyclic graphs $\mathcal{P}(N, E)$. We denote $n_\rightarrow(e), n_\leftarrow(e)$, incoming and outgoing node of an edge $e \in E$, and $E_\rightarrow(n), E_\leftarrow(n)$ the sets of incoming and outgoing edges of a node $n \in N$. The core element of (Multi-Dimensional) LSTMs is the LSTM Cell $c_n$ computed as:

$$c_n = \sum_{n' \in \mathcal{P}_\mathcal{G}(n)} f_{nn'} \odot c_{n'} + i_n \odot z_n \tag{1}$$

$$h_n = o_n \odot \tanh(c_n) \tag{2}$$

with hidden state / output $h_n$ and $\mathcal{P}_\mathcal{G}(n)$ denoting the parents of node $n$ in a DAG $\mathcal{G}$. In standard LSTMs, the input $i_n$, forget $f_n$, output gates $o_n$ and the Cell update $z_n$ are dependent on external inputs $x_n$ and on the previous/parent hidden states $(h_{n'})_{n' \in \mathcal{P}_\mathcal{G}(n)}$, which makes them non-linear. In contrast, for undirected graphs, DAG LSTMs can be applied by ordering the nodes first, thereby fixing edge directions and using a DAG LSTM bi-directionally. Note that there is typically more than one ordering option, where different orders can lead to different outcomes.

### 3.2 Linear RNNs

Linear RNNs resemble the structure of the original LSTM. For linearization, they remove the previous hidden state dependency of the gates and cell update. Additionally, they commonly include a state expansion dimension or query / key dimension. This enables a form of associative memory with inner-product retrieval in relation to quadratic energy-based models [Hopfield, 1982, Schmidhuber, 1992]. To include all relevant dimensions, we switch to the Einstein sum convention notation (so dimensions are visible in the indices) and absorb the input gate in the key and the output gate in the query. In reverse, query and key can also be interpreted as extended input and output gates. Additional normalization, as in mLSTM [Beck et al., 2025b], can be interpreted as a parallel execution with similar inputs and reduced dimensionality.

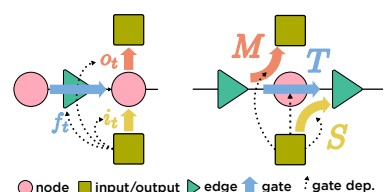

○ node  ☐ input/output  ▷ edge  ⬆ gate  ⁝ gate dep.

**Figure 3:** Transition from input/-forget/output gating in between nodes (and input/output) in (linear) RNNs towards **Source/Transition/-Mark gating** between edges and input/output (bottom/top) in pLSTMs. As for linear RNNs, the gates do not depend on states, but inputs only.

$$C_{tkv} = F_t C_{t-1kv} + K_{tk} V_{tv} \tag{3}$$
$$H_{tv} = Q_{tk} C_{tkv} \tag{4}$$

For certain architectures, the scalar forget gate $F_t$ is extended to the key dimension as $F_{tk}$ in GLA [Yang et al., 2023] or a matrix in DeltaNet as $F_{tkk'} C_{t-1k'v}$ with $F_{tkk'} = 1 - \beta_t K_{tk} K_{tk'}$ [Schlag et al., 2021, Yang et al., 2024] or Gated DeltaNet [Yang et al., 2025]. DeltaProduct [Siems et al., 2025] uses a product of multiple of these Householder-matrices. The important part is the linearization, by removing the dependence on the previous hidden state. All of $F_t$, $K_{tk}$, $V_{tv}$ and $Q_{tk}$ depend only on the input of the layer $x_{td}$ for this time step. This structure enables a chunkwise-recurrent form for efficient execution on modern hardware [Beck et al., 2025a, Yang et al., 2023, 2024].

# 4 pLSTM

Similar to how LSTMs are extended to general DAGs, this can be applied to linear RNNs. Here, we extend the setting of MDRNNs and DAG LSTMs to enable guided directional propagation. Instead of being associated with a node in the graph, the Cell state $C_e$ is now associated with an edge. The inputs and outputs / hidden states are still associated with nodes. Using this reformulation, we can define generalized linear DAG networks that resemble the LSTM structure (see Figures 3 and 6):

$$C_{ekv} = \sum_{e' \in E_\rightarrow(n_\rightarrow(e))} T_{ee'}^{n_\rightarrow(e)} C_{e'kv} + K_k^{n_\rightarrow(e)} V_v^{n_\rightarrow(e)} S_e^{n_\rightarrow(e)} \tag{5}$$

$$H_v^n = Q_k^n \sum_{e \in E_\rightarrow(n)} M_e^n C_{ekv} + K_k^n V_v^n D^n Q_k^n \tag{6}$$

The Cell state receives an external input via the Source $S_e^{n_\rightarrow(e)}$ at the incoming node in addition to a sum of the Cell states of incoming edges to the incoming node modified by the respective Transition $T_{ee'}^{n_\rightarrow(e)}$. The hidden state / network output $H_v^n$ at node $n$ is a sum of all incoming edge Cell states $C_{ekv}$ modified by Mark $M_e^n$ and projected by query $Q_k^n$. Note that the Source, Transition, and Mark gates are influenced by the input at node $n$, and optionally by the edge features of their associated local edges. Functionally, the Source replaces the input gate, the Transition the forget gate, and the Mark the output gate of a traditional LSTM [Hochreiter and Schmidhuber, 1997, Gers, 1999], while acting here on the node-edge, edge-edge, and edge-node combinations. The direct $D^n$ term represents a skip directly from input to output at a node and also integrates the $Q_k^n, K_k^n, V_v^n$ terms. These equations define general parallelizable Linear Source Transition Mark (pLSTM) networks on DAGs. This S-T-M structure can be computed sequentially, going through the DAG in a topological node order.

## 4.1 Parallelization on DAGs

### 4.1.1 Naïve Parallelization over Paths

Since all operations are linear, the iterative definition of Equation 5 can also be resolved to a sum over all ancestor nodes and a combination of Source, path Transition and Mark over all paths $\mathcal{P}(e', e)$ connecting nodes $n'$ and $n$ via first edge $e'$ and last edge $e$:

$$H_v^n = \sum_{n' < n} K_k^{n'} V_v^{n'} \underbrace{\sum_{e', e} S_{e'}^{n'} \left( \sum_{P \in \mathcal{P}(e', e)} \prod_{n^p \in P} T_{e_\leftarrow^{n^p} e_\rightarrow^{n^p}}^{n^p} \right) M_e^n}_{G^{n'n}} Q_k^n + K_k^n V_v^n \underbrace{D^n}_{G^{nn}} Q_k^n \tag{7}$$

The middle STM part in Equation 7 can be precomputed into the resulting gating matrix $G^{n'n}$, and the state expansion and projection via $Q_k^n$ and $K_k^{n'}$ can be integrated externally, similar to classical (linear) self-attention [Vaswani et al., 2017, Katharopoulos et al., 2020], the $D^n$ term represents the diagonal entries. Computing the gating matrix via all paths is infeasible due to their exponential number (see Appendix D.1).

### 4.1.2 Hierarchical Parallelization

Similar to an associative scan in 1D, we can apply a divide-and-conquer approach for hierarchical parallelization. Recursively, we merge Source-Transition, Transition-Transition, and Transition-Mark combinations to higher-order Source, Transition, and Mark objects that cover more than a single node. In Appendix B.2, we show how this procedure works for DAGs. For regular grids such as sequences in 1D and images in 2D, we can derive a scheme of einsum, concatenation, and padding operations, which are well supported by modern hardware and parallel computation frameworks. In Appendix B.3, we show the parallelization in 1D. In Appendix B.4, we show the parallelization scheme for 2D grids. To better convey the core idea of pLSTM parallelization, we include a sketch for all three cases and pLSTM in general in Figure 6 of the appendix.

## 4.2 Long-Range Stability

For sequential data, it is known since the seminal LSTM work of Hochreiter and Schmidhuber [1997] that having forget gates larger or smaller than one in magnitude leads to exploding/vanishing activations and gradients. For general DAGs, this was not explored yet, with recent analyses on undirected GNNs [Arroyo et al., 2025]. In DAG LSTM, Tree LSTM, Grid LSTM, or MD LSTM, potentially exploding parts are limited only in the hidden state by non-linear, bounded activation functions (i.e., tanh) [Hochreiter and Schmidhuber, 1997]. The Cell states, however, can grow exponentially if the forget gates are limited to only one. See Appendix D.1 for details.

Using the more general setting of Transitions $T_{ee'}^n$ from edge $e'$ to edge $e$, as introduced in this work, we can derive two modes of stability. First, a propagation mode (**P-mode**) that covers all paths in the DAG, but is effectively limited to a line as mode of propagation. Second, a distribution mode (**D-mode**) that limits the Transitions to a subset, such that their resulting line-graph is reduced from a DAG to a multi-tree Jung [1978].

The line graph of a DAG $\mathcal{G} = (N, E)$ is the DAG $\tilde{\mathcal{G}} = (E, \tilde{E})$ formed by the edges $E$ of the original DAG and the line edges $\tilde{e} \in \tilde{E} \iff \tilde{e} = (e_1, e_2), \exists n \in N : e_1 \in E_{\rightarrow}(n) \cap e_2 \in E_{\leftarrow}(n)$, which connect two original edges if these are connected by a node in $\mathcal{G}$ [Whitney, 1932]. Given the previous definitions of Cell states $C_e$ and Transitions $T_{ee'}^n$, these can be interpreted as states and edges on the nodes of the line graph $\tilde{\mathcal{G}}$. The Transitions $T_{ee'}$ can be viewed as entries of the adjacency matrix $\boldsymbol{T}$ of $\tilde{\mathcal{G}}$. This way, we can apply the theory of power iterations and matrix norms. The application of a pLSTM of Equation 5 is equivalent to the application of the power iteration and can be bounded by compatible matrix norms $|\cdot|$:

$$|\boldsymbol{c}| = \left| \left( \sum_{p=0}^{\infty} \boldsymbol{T}^p \right) \boldsymbol{s} \right| = \left| \left( \sum_{p=0}^{P_{\tilde{\mathcal{G}}}} \boldsymbol{T}^p \right) \boldsymbol{s} \right| \stackrel{|\boldsymbol{T}| \leq 1}{\leq} P_{\tilde{\mathcal{G}}} |\boldsymbol{s}| \tag{8}$$

As $\mathcal{G}$ is a DAG, also its line graph $\tilde{\mathcal{G}}$ is a DAG and this one's adjacency matrix $T_{ee'}$ is nilpotent, for a certain power $\exists P_{\tilde{\mathcal{G}}} \leq |N| : \forall p > P_{\tilde{\mathcal{G}}} : \boldsymbol{T}^p = \boldsymbol{0}$. For simplicity, we use matrix-vector notation here for Cell states $\boldsymbol{c}$, Transitions $\boldsymbol{T}$, and Sources $\boldsymbol{s}$ - without key-value extension.

The first stabilization option for the **P-mode** is to limit the norm of $\boldsymbol{T}$ as in Equation 8. A node-local option to achieve this is to limit the Transition entries $T_{ee'}^n$ per node as:

$$\sum_{e \in E_{\leftarrow}(n)} |T_{ee'}^n| \stackrel{!}{\leq} 1 \tag{9}$$

This limits the respective column $e'$ in the line graph adjacency matrix $\boldsymbol{T}$, and in turn limits its $L_1$-norm $||\boldsymbol{T}||_1$. Since the $L_1$-norm is sub-multiplicative [Horn and Johnson, 1985], this can be applied to the matrix powers in Equation 8 and keeps the $L_1$-norms of the Cell states $C_e$ limited. Keeping the norm fixed at one exactly enables long-range propagation (e.g., by division of the column entries by the norm). This stabilizes the gradient propagation with an $L_\infty$ bound, as gradients are propagated backwards and the applied transposed adjacency matrix $\boldsymbol{T}^T$ is row-(sub)-normalized and therefore $L_\infty$ bounded (the dual norm to the $L_1$ norm).

For the second stabilization option, the **D-mode**, we do not limit a matrix norm of $\boldsymbol{T}$, but instead reduce $\boldsymbol{T}$ or $\tilde{\mathcal{G}}$ from a DAG to a multitree [Jung, 1978], i.e., a directed graph, such that two nodes are only connected by a single path. The original graph $\mathcal{G}$ is not reduced to such a multitree by that. Since there is only one path in the line graph structure, there is only one path term in Equation 7, so there is no exponential explosion of path numbers, and limiting single $\boldsymbol{T}$ entries by one is sufficient.

## 4.3 pLSTM on regular grids

Although pLSTMs can be used and parallelized on general DAGs, the additional structure greatly helps with the parallelization for regular grids. All operations can be decomposed into generalized matrix operations in the form of views, einsums, concatenations, and paddings. We show this for the 1D case of sequences in Appendix B.3.

### 4.3.1  pLSTM in 2D - images

On a 2D undirected grid, pLSTM can be applied in different ways since there are more than two options for DAGs covering the undirected graph. There are, however, four distinct DAG covers, for which the local structure translates to a global structure: allowing only Transitions exclusively left or right and up or down, both in combination: $\nearrow, \searrow, \nwarrow, \swarrow$. We focus on the first, the right-down combination, for the description, but all four combinations should be used to cover all directional interactions. Note that here, nodes correspond to pixels/patches and edges to the respective connections between them, including the introduced direction. For the parallelization, we derive a hierarchy of different levels of Source, Transition and Mark, covering larger chunks of multiple nodes. This is also depicted in Figure 6 and described in Appendix Section B.2 and B.4.

As Transitions follow all edge combinations, there are four options again: $\rightarrow, \searrow, \searrow, \downarrow$. The level zero Source, Transition, and Mark tensors are therefore: $S_{xy}^{\rightarrow}, S_{xy}^{\downarrow}, T_{xy}^{\rightarrow}, T_{xy}^{\searrow}, T_{xy}^{\searrow}, T_{xy}^{\downarrow}, M_{xy}^{\rightarrow}, M_{xy}^{\downarrow}$. The general DAG is depicted in Figure 6 in the center right. As the in- and out-degrees of this DAG are now larger than one, there are more stabilization options, namely the P-mode and D-mode (Section 4.2).

**P-mode**  When we allow all Transition options $\rightarrow, \searrow, \searrow, \downarrow$, we are in the P-mode. In general, the Transition matrix at position $x, y$ looks like $T_{xy} = \begin{bmatrix} T_{xy}^{\rightarrow} & T_{xy}^{\searrow} \\ T_{xy}^{\searrow} & T_{xy}^{\downarrow} \end{bmatrix}$. The P-mode stabilization now limits this matrix to absolute column sums $\leq 1$. This limits the structure at criticality to $\begin{bmatrix} \alpha & \beta \\ 1 - \alpha & 1 - \beta \end{bmatrix}$. When setting $\beta = \alpha$, we even arrive at a geometrical interpretation of this Transition: it directionally propagates the signal with maximal amplitude along the line defined by $\frac{\Delta x}{\Delta y} = \alpha$ (see Appendix D.2 for the derivation). In Figure 6, bottom right, we depict the receptive field of a single non-zero Source with constant Transitions of this structure across the grid. For a practical parameterization, we, therefore, fix this structure with the 'angle' of propagation $\alpha$ and an additional decay factor $\gamma$:

$$T = \gamma \begin{bmatrix} \alpha & \alpha \\ (1 - \alpha) & (1 - \alpha) \end{bmatrix} \tag{10}$$

**D-mode**  While the P-mode offers a directional propagation, the receptive field of a node/patch/pixel is limited to roughly a line, it cannot reach all other nodes equally. The D-mode can provide this by losing the notion of directionality. For the D-mode, either $T_{xy}^{\searrow}$ or $T_{xy}^{\searrow}$ have to be set to zero globally. In this way, two edges can only be connected via a single path. In Figure 6, the bottom right, the second option is shown, with an additional decay. At criticality, this node can still reach all other nodes to its bottom right with magnitude one, a long-range propagation in two dimensions.

**Directional Initialization and Combination**  Using the concept of multiple heads, as used in Vaswani et al. [2017], we can initialize our network to cover multiple directions in P-mode at criticality on initialization. Similarly, we can initialize different decay scales for both P- and D-mode for different heads. By using the P- and D-mode in alternating layers, we leverage both their potential.

**State-Tracking Expansion**  In Appendix B.5, we show how pLSTMs can be extended for state tracking via non-diagonal Transitions [Merrill et al., 2024, Siems et al., 2025, Grazzi et al., 2025].

## 5  Experiments

We first showcase the theoretical advantages of pLSTM on the synthetic arrow-pointing extrapolation task (see Section 5.1). Then we highlight the benefits of pLSTM for two-dimensional input data on ImageNet-1K [Deng et al., 2009, Russakovsky et al., 2015], demonstrating scalability to large-scale datasets (see Sections 5.2 and 5.3). Finally, we illustrate how pLSTM scales to more than two input dimensions on established graph benchmarks (see Section 5.4).

### 5.1  Arrow Pointing Extrapolation

In this task, an image containing an arrow and a circle must be classified as to whether the arrow is pointing to the circle (see also Figure 2). This is a long-range directional task as it involves the relative

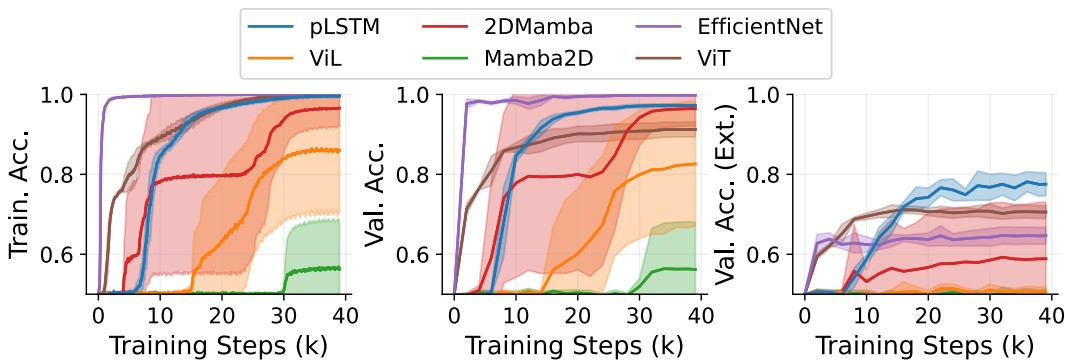

**Figure 4:** Training curves for the **Arrow Pointing Extrapolation task**, averaged over 5 seeds with 90% CI. ViT and EfficientNet can quickly match the training set (left), and EfficientNet reaches the best validation performance of all models on the samples of the same resolution as seen during training. pLSTM performs best on the extrapolation set (right) by a significant gap. While Mamba2D and 2DMamba should cover restricted modes of pLSTM, their learning shows high variance. ViL and Mamba2D do not extrapolate at all beyond random performance.

positioning of two objects, in conjunction with local information from one of them - the direction of the arrow. To test for arrow pointing extrapolation capabilities, we generate a balanced dataset of 100k arrow pointing images at resolution $192 \times 192$, with positions of the arrow and circle randomly sampled. For validation, we generate 5120 images in the same resolution and at resolution $384 \times 384$ - to test for extrapolation capabilities. For all models, including the ViT [Dosovitskiy et al., 2021] baseline, we use a bicubic interpolation of the positional embeddings (via `jax.image.resize`) for scaling to higher resolutions (see also [Dosovitskiy et al., 2021]). For details on the training, see Appendix E.2. In Figure 4, we show the learning curves for pLSTM and all baselines. We find that pLSTM performs well on both the training and validation tasks. Importantly, pLSTM exhibits the strongest extrapolation abilities. In contrast, ViL performs poorly because it traverses the image patches sequentially along the scanline form and misses out on important directional information. Despite its global receptive field, ViT falls short on extrapolation to larger image resolutions, which we surmise is due the its positional encoding. Similarly, EfficientNet exhibits strong performance on training/validation tasks, but fails to extrapolate. Finally, pLSTM considerably outperforms Mamba2D (which covers the P-mode) and 2DMamba (which covers the D-mode), despite their multidimensional nature. Also note that, in practice, pLSTM is much faster than Mamba2D or 2DMamba on real hardware, despite missing kernels, and on par with ViL (see 6). In Figure 5, we highlight the performance differences between D- and P-mode-only models on the arrow pointing task.

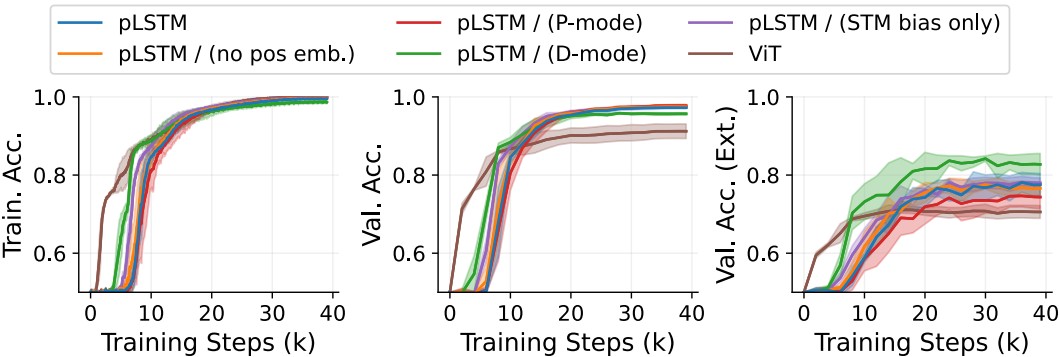

**Figure 5:** Training curves for the **Arrow Pointing Extrapolation task**, averaged over 5 seeds with 90% CI on different model ablations. P-mode by itself performs worse, D-mode by itself is not as general in interpolation, but performs better on extrapolation compared to other models. pLSTM does not rely on the positional embedding for strong performance.

**Table 1: ImageNet-1K** pre-training accuracy. All models use a patch size of $16 \times 16$ with $224 \times 224$ resolution at maximum. Models with "+" in their "Epochs" column pre-train on lower resolution, followed by fine-tuning on $224 \times 224$ resolution for some epochs. Values of reference models are taken from Alkin et al. [2025] and the original work for 2DMamba, Mamba2D and EfficientNet [Tan and Le, 2019, Zhang et al., 2024b, Baty et al., 2024]. Note that EfficientNet, as a CNN-based baseline, was also trained on larger resolutions for the scaled-up versions. Due to the chunkwise-recurrent option, pLSTM FLOPs could be optimized further (see Appendix B.7).

| Model | Epochs | #Params | FLOPS | IN-1K |
|---|---|---|---|---|
| EfficientNet-B0 [Tan and Le, 2019] | ? | 5M | 0.39G | 77.1 |
| DeiT-T [Touvron et al., 2021] | 300 | 6M | 1.3G | 72.2 |
| DeiT-III-T (reimpl.) [Touvron et al., 2022] | 800+20 | 6M | 1.1G | 75.4 |
| VRWKV-T [Duan et al., 2024] | 300 | 6M | 1.2G | 75.1 |
| Vim-T [Zhu et al., 2024] | 300 | 7M | 1.5G | 76.1 |
| ViL-T [Alkin et al., 2025] | 800+20 | 6M | 1.3G | **78.3** |
| pLSTM-Vis-T | 800+20 | 6M | 1.4G | 75.2 |
| EfficientNet-B4 [Tan and Le, 2019] | ? | 19M | 1.8G | **82.9** |
| DeiT-S [Touvron et al., 2021] | 300 | 22M | 4.6G | 79.8 |
| DeiT-III-S (reimpl.) [Touvron et al., 2022] | 400+20 | 22M | 4.6G | 80.3 |
| ConvNeXt-S (*iso.*) [Liu et al., 2022] | 300 | 22M | 4.3G | 79.7 |
| VRWKV-S [Duan et al., 2024] | 300 | 24M | 4.6G | 80.1 |
| Vim-S [Zhu et al., 2024] | 300 | 26M | 5.3G | 80.5 |
| Mamba2D-T [Baty et al., 2024] | 300 | 27M | - | 82.4 |
| 2DMamba-T [Zhang et al., 2024a] | ? | 30M | 4.9G | 82.8 |
| ViL-S [Alkin et al., 2025] | 400+20 | 23M | 4.7G | 81.5 |
| pLSTM-Vis-S | 400+20 | 23M | 4.9G | 80.7 |
| EfficientNet-B6 [Tan and Le, 2019] | ? | 43M | 19G | **84.0** |
| DeiT-B [Touvron et al., 2021] | 300 | 86M | 17.6G | 81.8 |
| DeiT-III-B (reimpl.) [Touvron et al., 2022] | 400+20 | 87M | 16.8G | 83.5 |
| ConvNeXt-B (*iso.*) [Liu et al., 2022] | 300 | 87M | 16.9G | 82.0 |
| VRWKV-B [Duan et al., 2024] | 300 | 94M | 18.2G | 82.0 |
| 2DMamba-S [Zhang et al., 2024a] | ? | 50M | 8.8G | 83.8 |
| ViL-B [Alkin et al., 2025] | 400+5 | 89M | 17.9G | 82.4 |
| pLSTM-Vis-B | 400+20 | 89M | 18.2 G | 82.5 |

## 5.2 ImageNet1k

To test for real-world image model capabilities, we train pLSTM using the schedule of DeiT-III [Touvron et al., 2022], comparing our architecture to other vision models. pLSTM performs similar to other popular approaches. With the integration of local features, e.g., including additional convolutions such as in Vision LSTM [Alkin et al., 2025], we see room to narrow the gap to SOTA. For a detailed discussion on the results and relation to previous work, see Appendix E.3.

## 5.3 ImageNet-1k Ablation

We ablate our pLSTM and compare against a ViT trained using the ViT-T model scale and a simpler single pre-training schedule close to DeiT [Touvron et al., 2021] at $224 \times 224$ resolution (see Appendix E.3.1) for details. At this scale, pLSTM outperforms ViT by a significant margin. Results are shown in Table 2.

---

[0] https://github.com/facebookresearch/fvcore

**Table 2: Ablation** of pLSTM variants and ViT (DeiT) re-training on **ImageNet-1k**.

| Model | ImageNet-1k (top-1) ↑ |
|---|---|
| pLSTM | **75.51** |
| pLSTM / (no posemb.) | 75.22 |
| pLSTM / (P-mode only) | 74.86 |
| pLSTM / (D-mode only) | 75.13 |
| pLSTM / (STM bias only) | 75.10 |
| ViT | 73.49 |

**Table 3:** Long-Range Graph Benchmark Results [Dwivedi et al., 2022]

| Model | Peptides -struct (MAE) ↓ | Peptides -func (Avg. Prec.) ↑ |
|---|---|---|
| GAT [Veličković et al., 2018] | 0.270 | 0.525 |
| GCN [Kipf and Welling, 2017] | 0.264 | 0.534 |
| GIN [Xu et al., 2018] | 0.328 | **0.593** |
| LSTM-GNN [Liang et al., 2016] | 0.274 | 0.502 |
| MPNN [Gilmer et al., 2017] | **0.260** | 0.557 |
| pLSTM | 0.264 | 0.536 |

## 5.4 Graphs

Small molecules and proteins are typically depicted as an undirected graph. However, both exist as 3D structures in the real world, and GNNs trained on such datasets might benefit from directional information propagation, as there is an underlying spatial relation. We test this hypothesis on popular small molecules and bioinformatics datasets from the TUDataset benchmark [Morris et al., 2020] and the Long-Range Graph Benchmark [Dwivedi et al., 2022]. The results in Tables 3 and 4 show that pLSTM can compete with popular GNN architectures on those datasets. Similar to the computer vision experiments (Section 5.2), pLSTM alternates between the P-mode and the D-mode. Since there is no external notion of direction, we use node and edge features to adapt the Transitions.

**Table 4:** Test accuracy on small molecule and bioinformatics datasets as provided by **TUDataset**.

| Model | MUTAG | NCI1 | PROTEINS | PTC_FM | AVG ↑ |
|---|---|---|---|---|---|
| GAT [Veličković et al., 2018] | 0.7822 ± 0.09 | 0.7968 ± 0.03 | 0.7215 ± 0.03 | 0.6105 ± 0.05 | 0.7277 ± 0.06 |
| GCN [Kipf and Welling, 2017] | 0.7234 ± 0.08 | 0.7852 ± 0.02 | 0.7395 ± 0.03 | **0.6162 ± 0.04** | 0.7161 ± 0.05 |
| GIN [Xu et al., 2018] | 0.8251 ± 0.10 | **0.8175 ± 0.02** | 0.7350 ± 0.04 | 0.6097 ± 0.10 | **0.7468 ± 0.08** |
| LSTM-GNN [Liang et al., 2016] | 0.7450 ± 0.11 | 0.7951 ± 0.02 | **0.7503 ± 0.04** | 0.6076 ± 0.04 | 0.7245 ± 0.06 |
| MPNN [Gilmer et al., 2017] | 0.7450 ± 0.09 | 0.8012 ± 0.02 | 0.7350 ± 0.04 | 0.5786 ± 0.07 | 0.7149 ± 0.06 |
| pLSTM | **0.8512 ± 0.06** | 0.7324 ± 0.03 | 0.7502 ± 0.05 | 0.6133 ± 0.08 | 0.7368 ± 0.06 |

## 6 Conclusion

In this work, we introduce pLSTM, which unites the benefits of MDRNNs [Graves et al., 2007] and the recently introduced xLSTM [Beck et al., 2025b]. pLSTM overcomes the limitations of modern recurrent architectures when applied to multi-dimensional data, such as images and graphs. To achieve this, we modify the gating structure of xLSTM and introduce Source, Transition, and Mark gates. Then, we derive a parallelization scheme that enables processing data in multiple dimensions concurrently. pLSTM comes with a P-mode and a D-mode, which together enable a large and auto-tunable effective receptive field. We demonstrate the theoretical advantages of pLSTM on the arrow-pointing task and highlight its ability to generalize to varying grid resolutions. Finally, we show the efficacy of pLSTM both on classical computer vision and graph benchmarks.

**Limitations & Future Work** pLSTM shows promising results across different domains, however, there is still room for improvement compared to highly optimized domain-specific models. With the incorporation of domain-specific inductive biases, we are positive that the results can be further improved. While pLSTMs should theoretically enable long-range propagation of activations and gradients in multiple dimensions for recurrent, subquadratic models, the arrow pointing extrapolation task only tests this in a restricted way. Moreover, while pLSTM models generalize better than ViTs, which use position embeddings to encode spatial relations, there is still a gap to perfect extrapolation. This leaves room for improvements, both on the data side of harder multi-dimensional long-range benchmarks and the architectural side to better generalize to that data. Given the flexibility of pLSTM to handle data with rich multi-dimensional structure, we anticipate that pLSTMs can be successfully applied across a broader spectrum of challenging domains, including biology, chemistry, medical imaging, and other scientific domains.

## Acknowledgments and Disclosure of Funding

We thank Benedikt Alkin, Pieter-Jan Hoedt, Phillip Lippe, Maximilian Beck, Kajetan Schweighofer, Erich Kobler, and Günter Klambauer for helpful discussions and feedback.

Korbinian Pöppel was supported by the Konrad Zuse School of Excellence in Learning and Intelligent Systems (ELIZA) through the DAAD program Konrad Zuse Schools of Excellence in Artificial Intelligence, sponsored by the Federal Ministry of Education and Research during his ELLIS visit at Uni Freiburg, where this work was finished.

The ELLIS Unit Linz, the LIT AI Lab, the Institute for Machine Learning, are supported by the Federal State Upper Austria. We thank the projects FWF AIRI FG 9-N (10.55776/FG9), AI4GreenHeatingGrids (FFG- 899943), Stars4Waters (HORIZON-CL6-2021-CLIMATE-01-01), FWF Bilateral Artificial Intelligence (10.55776/COE12). We thank NXAI GmbH, Audi AG, Silicon Austria Labs (SAL), Merck Healthcare KGaA, GLS (Univ. Waterloo), TÜV Holding GmbH, Software Competence Center Hagenberg GmbH, dSPACE GmbH, TRUMPF SE + Co. KG.

We acknowledge EuroHPC Joint Undertaking for awarding us access to Leonardo at CINECA, Italy, and MareNostrum5 at BSC, Spain.

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

# A  Method Visualization

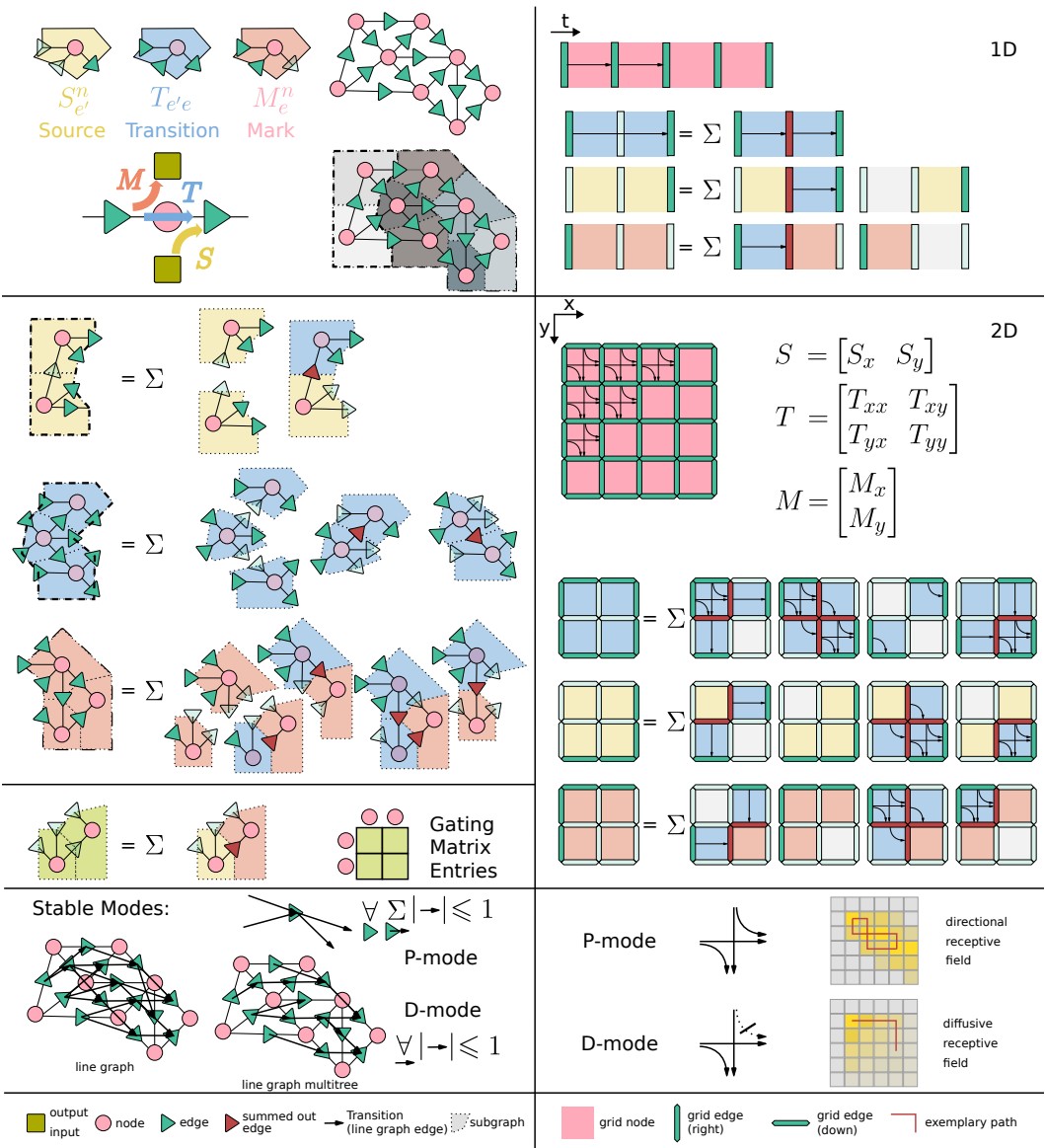

**Figure 6:** Illustration of **pLSTM** on general DAGs (left), and on 1D / 2D grids (right): In the top-left part, a general DAG is visualized, with Sources $S_{e'}^n$, Transitions $T_{e'e}$, and Marks $M_e^n$ populating node-outgoing-edge, incoming-edge-outgoing-edge, and incoming-edge-node pairs. Due to the associative structure of these linear operators, they can be combined; some examples are shown in the center left. The end-to-end (node-to-node) result is the Gating matrix. In the bottom left, the two stable modes are depicted: The P-mode, where the sum over absolute Transitions of an in-coming edge have to be limited by one, and the D-mode, where the line graph over edges is reduced to a multitree, requiring the remaining Transitions each to be limited in absolute to one. On the right, the application of this to regular 1D and 2D grids is shown. For 2D, the P-mode results in a directional propagation, whereas the D-mode results in a diffusive / un-directed distribution (shown with decay here).

## B  Method Details

### B.1  Loosely self-similar graphs

Graphs modeling hierarchical order or spatial extensions, as in meshes or grids do typically have low node degree and can be decomposed recursively into subgraphs of similar structure. We use the term loosely self-similar graphs for these. In particular, a graph $\mathcal{G} := \mathcal{G}^{(L)}$ should be decomposable into $M$ subgraphs $\{\mathcal{G}_a^{(L-1)} = (N_a^{(L-1)}, E_a^{(L-1)})\}_{a \in \{1..M\}}$, and boundary edges $B_{ab}^{(L-1)} \subset N_a^{(L-1)} \times N_b^{(L-1)}$ such that $\mathcal{G} = \mathcal{G}^{(L)} = \left( \bigcup_a N_a^{(L-1)}, \bigcup_{a,b} B_{ab}^{(L-1)} \cup \bigcup_a E_a^{(L-1)} \right)$. This decomposition should be applicable recursively down to single nodes, such that $|N_\alpha^{(0)}| = 1 \quad \forall \alpha$ with $\alpha = (a_{L-1}...a_0)$ the multi-index of successive decompositions. Ideally, all decompositions should be balanced, $|N_\alpha^{(l)}| - |N_\beta^{(l)}| \in \{-1, 0, 1\} \ \forall l, \alpha, \beta$, for optimal parallelization - which should serve as the definition of loosely self-similar. Choosing arbitrary subsets of nodes for subgraphs, one can always arrive at a decomposition of any graph, but the edges across subgraphs might not be balanced then.

### B.2  Hierarchical Parallelization

Given a decomposition of a (loosely self-similar) DAG, we can apply a divide-and-conquer strategy to the expensive Transition sum/product combination. We want to calculate now the $G^{n'n}$ matrix entry for two nodes $n', n \in N; n' \le n$, such that, at level $l+1$, both are in the subgraph $\mathcal{G}_\alpha^{l+1} : n', n \in N_\alpha^{l+1}$ but going one level lower, they are in distinct subgraphs $n' \in N_{\alpha:a}^l, n \in N_{\alpha:b}^l \ a \neq b$. With incoming and outgoing boundary edges $e' \in B_{\to\alpha:a}^l, e \in B_{\leftarrow\alpha:a}^l$, we define the generalized Source, Transition, and Mark tensors at level $l$:

$$S_e^{n',l,\alpha:a} := \sum_{e' \in E_\leftarrow(n')} S_{e'}^{n'} \sum_{P \in \mathcal{P}(e',e)} \prod_{n^P \in P} T_{e_\to^{n^P} e_\leftarrow^{n^P}}^n \tag{11}$$

$$T_{e'e}^{l,\alpha:k} := \sum_{P \in \mathcal{P}(e',e)} \prod_{n^P \in P} T_{e_\to^{n^P} e_\leftarrow^{n^P}}^n \tag{12}$$

$$M_{e'}^{n,l,\alpha:b} := \sum_{e \in E_\to(n)} \left( \sum_{P \in \mathcal{P}(e',e)} \prod_{n^P \in P} T_{e_\to^{n^P} e_\leftarrow^{n^P}}^n \right) M_e^n \tag{13}$$

Given these, the gating matrix entry can be calculated as:

$$G^{n'n} = \sum_{e \in B_{\leftarrow\alpha:a}^l} \sum_{e' \in B_{\to\alpha:j}^l} S_e^{n',l,\alpha:a} \left( \sum_{P^l \in \mathcal{P}^l(e',e)} \prod_{c \in P^l} T_{e_{\to m_c^l}^{l,\alpha:c} e_{\leftarrow m_c^l}^{l,\alpha:c}}^{l,\alpha:c} \right) M_{e'}^{n',l,\alpha:b}$$

Here, $\mathcal{P}^l$ is the set of "meta-paths" that connect two edges $e', e$ via subgraphs at level $l$, in these meta-paths the subgraphs act as "nodes" and the boundary edges $B_{\leftarrow\alpha:a}^l, B_{\to\alpha:a}^l$ as "edges". Note that, by Einstein sum convention we implicitly sum over multiple in-coming and out-going edges $e_{\to m_c^l}^{l,\alpha:c}$ and $e_{\leftarrow m_c^l}^{l,\alpha:c}$ for generalized Transitions $T^{l,\alpha:c}$ in $G^{n'n}$ and in the following. See Figure 6 for an illustration of this decomposition. In reverse, this means that higher-level Source, Transition, and Mark can be constructed recursively bottom-up from one level lower ($a$ chosen such that $n' \in N_{\alpha:a}^l$ or $n \in N_{\alpha:a}^l$):

$$S_e^{n',l+1,\alpha} = \sum_{e' \in B_{\leftarrow\alpha:a}^l} S_{e'}^{n',l,\alpha:a} \sum_{P^l \in \mathcal{P}^l(e',e)} \prod_{b \in P^l} T_{e_{\to m_b^l}^{l,\alpha:b} e_{\leftarrow m_b^l}^{l,\alpha:b}}^{l,\alpha:b} \tag{14}$$

$$T_{e'e}^{l+1,\alpha} = \sum_{P^l \in \mathcal{P}^l(e',e)} \prod_{b \in P^l} T_{e_{\to m_b^l}^{l,\alpha:b} e_{\leftarrow m_b^l}^{l,\alpha:b}}^{l,\alpha:b} \tag{15}$$

$$M_{e'}^{n,l+1,\alpha} = \sum_{e \in B_{\to\alpha:a}^l} \left( \sum_{P^l \in \mathcal{P}^l(e',e)} \prod_{b \in P^l} T_{e_{\to m_b^l}^{l,\alpha:b} e_{\leftarrow m_j^l}^{l,\alpha:b}}^{l,\alpha:b} \right) M_e^{n,l,\alpha:a} \tag{16}$$

Note that using this hierarchy only up to a certain level, one can optimize the parallelization to the used hardware. Whereas first, all the Source, Transition, Mark, and Gating matrix objects are constructed in parallel up to a certain level, the recurrent mode of Equation 5 is applied with generalized Source, Transition and Mark matrices of level $l$, in the topological ordering of the subgraphs at this level. This resembles the structure of the chunk-wise parallel formulation for linear RNNs.

## B.3 Parallelization of pLSTM in 1D - sequences

For a sequence as a DAG, pLSTMs reduce to linear RNNs. Here, all nodes and subgraphs have a single incoming and outgoing (boundary) edge. For parallelization, the decomposition is done hierarchically in powers-of-two splits. The $S, T, M$ and $G$ tensors are constructed as follows:

$$S_{an}^l = \left[ S_{(2a)(n\rfloor_0)}^{l-1} T_{(2a+1)}^{l-1}, S_{(2a+1)(n\rfloor_0)}^{l-1} \right]_0 \tag{17}$$

$$T_a^l = T_{(2a)}^{l-1} T_{(2a+1)}^{l-1} \tag{18}$$

$$M_{an}^l = \left[ M_{(2a+1)(n\rfloor_0)}^{l-1}, T_{(2a)}^{l-1} M_{(2a+1)(n\rfloor_0)}^{l-1} \right]_0 \tag{19}$$

$$G_{an'n}^l = \left[ \left[ G_{(2a)(n'\rfloor_0)(n\rfloor_1)}^l, 0 \right]_0 , \right. \tag{20}$$
$$\left. \left[ S_{(2a)(n'\rfloor_0)}^{l-1} M_{(2a+1)(n\rfloor_1)}^{l-1}, G_{(2a+1)(n'\rfloor_0)(n\rfloor_1)}^l \right]_0 \right]_1$$

with $a$ the external index (Source index $\in \{0..S/2^l - 1\}$ of level l), $n$ the internal index (node index $\in \{0..2^l - 1\}$ within generalized Source $S_a^l$). Here, $[]_i$ denotes a concatenation along the axis $i$, and $\rfloor_i$ and $\rceil_i$ denote these indices are used as axis $i$ for this concatenation - one taking the lower half, one the upper half, e.g. for vectors $a \in \mathbb{R}^N$; $b, c \in \mathbb{R}^{N/2}$:

$$a_n = \left[ b_{(n\rfloor_0)}, c_{(n\rfloor_0)} \right]_0 := \begin{cases} b_n & n < N/2 \\ c_{n-N/2} & n \geq N/2 \end{cases} \tag{21}$$

Note that we use a causal setting here. For a non-causal structure, one would apply this bi–bidirectionally, effectively filling the top-right entries of the gating matrix with a different Source/Mark combination for the opposite order. Equivalently, both directions can be added in their outputs while computing the parallel parts twice.

## B.4 Parallelization of pLSTM in 2D - images

Here, we now use $u$ for internal indices along the $x$-axis, and $v$ for internal indices along the $y$-axis, and $a, b, c$ for an index along the edge/boundary direction, while keeping $x, y$ for outer indices. These are needed for higher-level tensors, leading to a recursion shown below. The Source, Transition and Mark tensors at level $l$ have the following structure: $S_{xyuva}^{l,\mapsto}, S_{xyuva}^{l,\updownarrow}, T_{xyab}^{l,\rightarrow}, T_{xyab}^{l,\searrow}, T_{xyab}^{l,\nearrow}, T_{xyab}^{l,\downarrow}, M_{xyuvb}^{l,\mapsto}, M_{xyuvb}^{l,\updownarrow}$ with $u, v, a, b \in \{0..2^l - 1\}$.

$$S_{xyuwa}^{l+1,\mapsto} = \left[ \left[ \left[ S_{(2x)(2y)(u\rfloor_0)(w\rfloor_1)b}^{l,\mapsto} T_{(2x+1)(2y)b(a\rfloor_2)}^{l,\rightarrow}, S_{(2x+1)(2y)(u\rfloor_0)(w\rfloor_1)(a\rfloor_2)}^{l,\mapsto} \right]_0, \left[ 0_{(u\rfloor_0)(w\rfloor_1)(a\rfloor_2)}, 0_{(u\rfloor_0)(w\rfloor_1)(a\rfloor_2)} \right]_0 \right]_1, \right.$$
$$\tag{22}$$

$$\left[ \left[ S_{(2x)(2y)(u\rfloor_0)(w\rfloor_1)b}^{l,\mapsto} T_{(2x+1)(2y)bc}^{l,\nearrow} T_{(2x+1)(2y+1)c(a\rfloor_2)}^{l,\searrow} + S_{(2x)(2y)(u\rfloor_0)(w\rfloor_1)b}^{l,\updownarrow} T_{(2x)(2y+1)bc}^{l,\searrow} T_{(2x+1)(2y+1)c(a\rfloor_2)}^{l,\rightarrow}, \right.$$

$$\left. S_{(2x+1)(2y)(u\rfloor_0)(w\rfloor_1)b}^{l,\updownarrow} T_{(2x+1)(2y+1)bc}^{l,\searrow} \right]_0, \left[ S_{(2x)(2y+1)(u\rfloor_0)(w\rfloor_1)b}^{l,\mapsto} T_{(2x+1)(2y+1)b(a\rfloor_2)}^{l,\rightarrow}, S_{(2x+1)(2y+1)(u\rfloor_0)(w\rfloor_1)(a\rfloor_2)}^{l,\mapsto} \right]_0 \right]_1 \right]_2$$

$$\tag{23}$$

$$S_{xyuwa}^{l+1,\updownarrow} = \Bigg[ \Big[ \big[ S_{(2x)(2y)(u\rfloor_0)(w\rfloor_1)b}^{l,\updownarrow} T_{(2x)(2y+1)b(a\rfloor_2)}^{l,\downarrow}, 0_{(u\rfloor_0)(w\rfloor_0)(a\rfloor_2)} \big]_0, \big[ S_{(2x)(2y+1)(u\rfloor_0)(w\rfloor_1)(a\rfloor_2)}^{l,\updownarrow}, 0_{(u\rfloor_0)(w\rfloor_0)(a\rfloor_2)} \big]_0 \Big]_1,$$

$$\text{(24)}$$

$$\Big[ \big[ S_{(2x)(2y)(u\rfloor_0)(w\rfloor_1)b}^{l,\mapsto} T_{(2x+1)(2y)bc}^{l,\searrow} T_{(2x+1)(2y+1)c(a\rfloor_2)}^{l\downarrow} + S_{(2x)(2y)(u\rfloor_0)(w\rfloor_1)b}^{l,\updownarrow} T_{(2x)(2y+1)bc}^{l,\hookrightarrow} T_{(2x+1)(2y+1)c(a\rfloor_2)}^{\searrow},$$

$$S_{(2x+1)(2y)(u\rfloor_0)(v\rfloor_1)b}^{l,\updownarrow} T_{(2x+1)(2y+1)b(a\rfloor_0)}^{l,\downarrow} \big]_0, \big[ S_{(2x)(2y+1)(u\rfloor_0)(w\rfloor_1)b}^{l,\mapsto} T_{(2x+1)(2y+1)b(a\rfloor_2)}^{l,\searrow}, S_{(2x+1)(2y+1)(u\rfloor_0)(v\rfloor_1)(a\rfloor_2)}^{l,\updownarrow} \big]_0 \Big]_1 \Bigg]_2$$

$$T_{xyab}^{l,\rightarrow} = \Big[ \big[ T_{(2x)(2y)(a\rfloor_0)c}^{l,\rightarrow} T_{(2x+1)(2y)c(b\rfloor_1)}^{l,\rightarrow}, 0_{(a\rfloor_0)(b\rfloor_1)} \big]_0, \tag{25}$$
$$\big[ T_{(2x)(2y)(a\rfloor_0 c)}^{l,\downarrow} T_{(2x)(2y+1)cd}^{l,\searrow} T_{(2x+1)(2y+1)d(b\rfloor_1)}^{l,\rightarrow} + T_{(2x)(2y)(a\rfloor_0)c}^{l,\rightarrow} T_{(2x+1)(2y)cd}^{l,\searrow} T_{(2x+1)(2y+1)d(b\rfloor_1)}^{l,\hookrightarrow},$$
$$T_{(2x)(2y+1)(a\rfloor_0)c}^{l,\rightarrow} T_{(2x+1)(2y+1)c(b\rfloor_1)}^{l,\rightarrow}, 0_{(a\rfloor_0)(b\rfloor_1)} \big]_0 \Big]_1$$

$$T_{xyab}^{l,\downarrow} = \Big[ \big[ T_{(2x)(2y)(a\rfloor_0)c}^{l,\downarrow} T_{(2x)(2y+1)c(b\rfloor_1)}^{l,\downarrow}, 0_{(a\rfloor_0)(b\rfloor_1)} \big]_0, \tag{26}$$
$$\big[ T_{(2x)(2y)(a\rfloor_0 c)}^{l,\downarrow} T_{(2x)(2y+1)cd}^{l,\searrow} T_{(2x+1)(2y+1)d(b\rfloor_1)}^{l,\searrow} + T_{(2x)(2y)(a\rfloor_0)c}^{l,\rightarrow} T_{(2x+1)(2y)cd}^{l,\searrow} T_{(2x+1)(2y+1)d(b\rfloor_1)}^{l,\downarrow},$$
$$T_{(2x+1)(2y)(a\rfloor_0)c}^{l,\downarrow} T_{(2x+1)(2y+1)c(b\rfloor_1)}^{l,\downarrow}, 0_{(a\rfloor_0)(b\rfloor_1)} \big]_0 \Big]_1$$

$$T_{xyab}^{l,\hookrightarrow} = \Big[ \big[ T_{(2x)(2y)(a\rfloor_0)c}^{l,\hookrightarrow} T_{(2x+1)(2y)c(b\rfloor_1)}^{l,\rightarrow}, T_{(2x+1)(2y)(a\rfloor_0)(b\rfloor_1)}^{l,\hookrightarrow} \big]_0, \tag{27}$$
$$T_{(2x)(2y)(a\rfloor_0)c}^{l,\hookrightarrow} T_{(2x+1)(2y)cd}^{l,\searrow} T_{(2x+1)(2y+1)d(b\rfloor_1)}^{l,\hookrightarrow} + T_{(2x)(2y)(a\rfloor_0)c}^{l,\downarrow} T_{(2x)(2y+1)cd}^{l,\searrow} T_{(2x+1)(2y+1)d(b\rfloor_1)}^{l,\rightarrow},$$
$$T_{(2x+1)(2y)(a\rfloor_0)c}^{l,\downarrow} T_{(2x+1)(2y+1)c(b\rfloor_1)}^{l,\hookrightarrow} \big]_0 \Big]_1$$

$$T_{xyab}^{l,\searrow} = \Big[ \big[ T_{(2x)(2y)(a\rfloor_0)c}^{l,\searrow} T_{(2x)(2y+1)c(b\rfloor_1)}^{l,\downarrow}, T_{(2x)(2y+1)(a\rfloor_0)(b\rfloor_1)}^{l,\searrow} \big]_0, \tag{28}$$
$$T_{(2x)(2y)(a\rfloor_0)c}^{l,\searrow} T_{(2x)(2y+1)cd}^{l,\hookrightarrow} T_{(2x+1)(2y+1)d(b\rfloor_1)}^{l,\searrow} + T_{(2x)(2y)(a\rfloor)_0 c}^{l,\rightarrow} T_{(2x+1)(2y)cd}^{l,\searrow} T_{(2x+1)(2y+1)d(b\rfloor_1)}^{l,\downarrow},$$
$$T_{(2x)(2y+1)(a\rfloor_0)c}^{l,\rightarrow} T_{(2x+1)(2y+1)c(b\rfloor_1)}^{l,\searrow} \big]_0 \Big]_1$$

$$M_{xyuva}^{l,\mapsto} = \Bigg[ \Big[ \big[ M_{(2x)(2y)(u\rfloor_0)(v\rfloor_1)(a\rfloor_2)}^{l,\mapsto}, T_{(2x)(2y)(a\rfloor_2)b}^{l,\rightarrow} M_{(2x+1)(2y)(u\rfloor_0)(v\rfloor_1)b}^{l,\mapsto} \big]_0, \big[ T_{(2x)(2y)(a\rfloor_2)b}^{l,\searrow} M_{(2x)(2y+1)(u\rfloor_0)(v\rfloor_1)b}^{l,\updownarrow},$$

$$\text{(29)}$$

$$T_{(2x)(2y)(a\rfloor_2)b}^{l,\searrow} T_{(2x)(2y+1)bc}^{l,\hookrightarrow} M_{(2x+1)(2y+1)(u\rfloor_0)(v\rfloor_1)c}^{l,\mapsto} + T_{(2x)(2y)(a\rfloor_2)b}^{l,\rightarrow} T_{(2x+1)(2y)bc}^{l,\searrow} M_{(2x+1)(2y+1)(u\rfloor_0)(v\rfloor_1)c}^{l,\updownarrow} \big]_0 \Big]_1,$$

$$\Big[ \big[ 0_{(u\rfloor_0)(v\rfloor_1)(a\rfloor_2)}, 0_{(u\rfloor_0)(v\rfloor_1)(a\rfloor_2)} \big]_0, \big[ M_{(2x)(2y+1)(u\rfloor_0)(v\rfloor_1)(a\rfloor_2)}^{l,\mapsto}, T_{(2x)(2y+1)(a\rfloor_2)b}^{l,\rightarrow} M_{(2x+1)(2y+1)(u\rfloor_0)(v\rfloor_1)b}^{l,\mapsto} \big]_0 \Big]_1 \Bigg]_2$$

$$M_{xyuva}^{l,\updownarrow} = \Bigg[ \Big[ \big[ M_{(2x)(2y)(u\rfloor_0)(v\rfloor_1)(a\rfloor_2)}^{l,\updownarrow}, T_{(2x)(2y)(a\rfloor_2)b}^{l,\hookrightarrow} M_{(2x+1)(2y)(u\rfloor_0)(v\rfloor_1)b}^{l,\mapsto} \big]_0, \big[ T_{(2x)(2y)(a\rfloor_2)b}^{l,\downarrow} M_{(2x)(2y+1)(u\rfloor_0)(v\rfloor_1)b}^{l,\updownarrow},$$

$$\text{(30)}$$

$$T_{(2x)(2y)(a\rfloor_2)b}^{l,\hookrightarrow} T_{(2x+1)(2y)bc}^{l,\searrow} M_{(2x+1)(2y+1)(u\rfloor_0)(v\rfloor_1)c}^{l,\updownarrow} + T_{(2x)(2y)(a\rfloor_2)b}^{l,\downarrow} T_{(2x)(2y+1)bc}^{l,\hookrightarrow} M_{(2x+1)(2y+1)(u\rfloor_0)(v\rfloor_1)c}^{l,\mapsto} \big]_0 \Big]_1,$$

$$\Big[ \big[ 0_{(u\rfloor_0)(v\rfloor_1)}, M_{(2x+1)(2y)(u\rfloor_0)(v\rfloor_1)}^{l,\updownarrow} \big]_0, \big[ 0_{(u\rfloor_0)(v\rfloor_1}, T_{(2x+1)(2y)(a\rfloor_2)b}^{l,\downarrow} M_{(2x+1)(2y+1)(u\rfloor_0)(v\rfloor_1)b}^{l,\updownarrow} \big]_0 \Big]_1 \Bigg]_2$$

## B.5  State-Tracking Extension

Recent research has shown that having scalar, positive Transitions $T^n_{e_\to e_\gets}$ does not allow for state tracking capabilities in these linear RNNs [Merrill et al., 2024, Grazzi et al., 2025]. We can generalize the Transition scalars to Transition matrices here as well: $T^n_{e^n_\to e^n_\gets} \to T^n_{e^n_\to e^n_\gets ij}\ i,j \in \{1..J_T\}$, $J_T$ being the Transition state tracking dimension. Note that in this way, the Source and Mark matrix have to include this extended dimension as well. We can define generalizations with additional state tracking dimensions $J_S, J_M$:

$$S^{n',l,\alpha}_e \to S^{n',l,\alpha}_{eki} \qquad\qquad k \in \{1..J_S\}, i \in \{1..J_T\} \qquad (31)$$

$$T^{l,\alpha}_{e'e} \to T^{l,\alpha}_{e'eij} \qquad\qquad i,j \in \{1..J_T\} \qquad (32)$$

$$M^{n,l+1,\alpha}_{e'} \to M^{n,l,\alpha}_{e'jm} \qquad\qquad j \in \{1..J_T\}, m \in \{1..J_M\} \qquad (33)$$

$$G^{n'n} \to G^{n'n}_{km} \qquad\qquad k \in \{1...J_S\}, m \in \{1..J_M\} \qquad (34)$$

This extension includes specific parameterizations such as defining the Transition matrix as $T_{ij} = 1 + \beta k_i k_j$ known from DeltaNet [Schlag et al., 2021]. For these, the chunk-wise parallelization of Yang et al. [2024] can be applied along multiple dimensions.

## B.6  Stability in the State Tracking Extension

The stabilization of Section B.5 can be applied in the state tracking extensions as well. In the extended case, the absolute values of Transitions are replaced by the spectral norm of the Transition matrix $(T_{e'eij})_{ij}$ defined by its largest singular value. The stability can be ensured by limiting the sum of the largest singular values (instead of scalar entries) for the P-mode or zero matrix entries, resulting in a line graph tree with maximally unit norm matrices for tree Transitions in the D-mode. There are several options to limit or set the Transition matrices by/to one in magnitude. A straightforward option is to parametrize not the Transition entries $T_{ij}$ directly, but to parametrize its singular value decomposition, with orthogonal (or unitary) matrices $\boldsymbol{T} = \boldsymbol{U\Sigma V}^T$. The singular values $\boldsymbol{\Sigma}$ can be limited in magnitude by a `tanh` (for multiplicative decay) or fixed to $\pm 1$ (long-range propagation). In the case of multiple directions (see 2D grid case), in the P-mode, they are multiplied by an additional softmax-limited propagation factor ("angle") - limiting the sum over the direction pairs in Equation 9. The orthogonal matrices $\boldsymbol{U}, \boldsymbol{V}$ can be parametrized by the product of Householder matrices (generated from vectors), or the exponential of the Lie-group / generating group of special orthogonal matrices: the skew-symmetric matrices (directly parameterized). With these parametrizations, in turn depending on the network inputs (at nodes), state-tracking capabilities can be achieved while maintaining long-range stability [Merrill et al., 2024, Grazzi et al., 2025].

## B.7  Chunkwise-Parallel Form

Given the hierarchical structure of the parallelization shown in Sections B.2, B.3, and B.4, one can stop at a certain level of it. At this level, the resulting objects will again form a higher-level DAG or grid, which can now be processed sequentially, in the topological ordering of this DAG. This way, the quadratic complexity introduced by the parallelization is only introduced up to the optimal level of the hardware's parallelization capabilities. While typically the FLOP-optimal point is at lower parallelization, Beck et al. [2025a] showed that a certain degree of parallelization is hardware optimal.

**Multi-Directional Form**  For the 2D grid case of images, we use pLSTM in all four direction combinations of the 2D DAG at once in parallel, at each node going down/right, down/left, up/right, up-left. By pre-computing all of them in full parallelization, we can add up their gating matrices, resulting in just a single pass of the gated linear attention computation (see Equation 7). This fully parallelized form was used for the vision experiments.

## C   Full Model Architecture

### C.1   pLSTM for Vision

For the pLSTM-Vis, we use the ViT [Dosovitskiy et al., 2021] with pre-LayerNorm (actually RMSNorm) structure as backbone architecture, replacing the Multi-Head Attention with a pLSTM layer. Note that in addition to query-, key-, value-, and out-projections, we have linear layers for Source, Transition, and Mark gates in the four direction combinations. In addition, we apply a multi-head RMSNorm. Note that the Source, Mark, and Direct gates are sigmoid-activated, and the Transition is tanh-activated. For details, we refer to the attached source code.

## D   Theoretical Analysis

### D.1   Exponential growth of path counts on a 2D grid

Assume a 2D grid structure of a DAG, then the number of paths between two nodes can be calculated in the following way. In total, the number of Transitions to the right and Transitions to the bottom are fixed by the position offsets $\Delta_x$ and $\Delta_y$ between the nodes. In total, the number of paths can be counted by the number of orderings of a string consisting of $\Delta_x$ times $\rightarrow$ and $\Delta_y$ times $\leftarrow$. This results in the combinatorial factor, which is exponential in the path length $\Delta_x + \Delta_y =: \Delta$:

$$\#\text{Paths} = \binom{\Delta_x + \Delta_y}{\Delta_x} = \frac{(\Delta_x + \Delta_y)!}{\Delta_x! \Delta_y!} \overset{\text{Stirling}}{\approx} \sqrt{\frac{(\Delta_x + \Delta_y)}{2\pi \Delta_x \Delta_y}} \frac{(\Delta_x + \Delta_y)^{\Delta_x + \Delta_y}}{\Delta_x^{\Delta_x} \Delta_y^{\Delta_y}}$$

$$\overset{\Delta_x := \beta \Delta}{=} \sqrt{\frac{1}{2\pi \Delta \beta (1 - \beta)}} \underbrace{\left( \beta^{-\beta} (1 - \beta)^{-(1-\beta)} \right)^{\Delta}}_{\geq 1} \tag{35}$$

Given Transitions / forget gates potentially all at 1, this exponential number of paths leads to an exponentially growing magnitude of the Cell state $C$, as of Equation 7. For a non-linear MD-RNN, like DAG-LSTM, this path count applies as well, assuming that all forget gates are fixed to one. Also in this case, the cell states accumulate all their ancestors' values via all paths.

### D.2   Long-Range Decay for P-mode in 2D

As the P-mode offers a directional propagation, here, we want to calculate the decay of one Source signal along the leading direction - given all Transitions are equal within the grid and at criticality: $T = \begin{bmatrix} \alpha & \alpha \\ (1 - \alpha) & (1 - \alpha) \end{bmatrix}$. This results in the following overall Transition from Source $S_{xy}$ at $x, y$ to Mark $M_{(x+\Delta_x)(y+\Delta_y)}$ at $x + \Delta_x, y + \Delta_y$ - with path length $\Delta = \Delta_x + \Delta_y \gg 1$, direction $\beta := \frac{\Delta_x}{\Delta}$:

$$T^{\text{full}}_{xy(x+\Delta_x)(y+\Delta_y)} = \sum_{\text{Paths}} T^{\Delta_x}_{r*} T^{\Delta_y}_{d*} = \binom{\Delta_x + \Delta_y}{\Delta_x} \alpha^{\Delta_x} (1 - \alpha)^{\Delta_y} = \binom{\Delta}{\beta \Delta} \alpha^{\beta \Delta} (1 - \alpha)^{(1-\beta)\Delta} \tag{36}$$

Notice the binomial structure of the equation that fixes the total sum $\sum_{\Delta_x \in \{0..\Delta\}} T^{\text{full}}_{xy(x+\Delta_x)(y+\Delta_y)}$ to 1, so the total activation is conserved along the diagonal. Differentiating this equation by $\beta$, we can get the direction of largest propagation:

$$0 \overset{!}{=} \partial_\beta \log T^{\text{full}}_{xy(x+\Delta_x)(y+\Delta_y)} = -\partial_\beta \left( \log \Gamma(\Delta + 1) + \log \Gamma(\beta \Delta + 1) + \log \Gamma((1 - \beta)\Delta + 1) \right) + \Delta \log(\alpha) - \Delta \log(1 - \alpha) \tag{37}$$

$$\overset{\text{Stirling}}{\approx} -\partial_\beta \left( \beta \Delta \log(\beta) + (1 - \beta)\Delta \log(1 - \beta) \right) + \Delta \log(\alpha) - \Delta \log(1 - \alpha)$$

$$= \Delta \log \left( \frac{\alpha (1 - \beta)}{\beta (1 - \alpha)} \right)$$

This implies $\beta = \alpha$ for the direction of largest propagation. Now, inserting this direction into the Transition product:

$$T^{\text{full}}_{xy(x+\Delta_x)(y+\Delta_y)}|_{\beta=\alpha} = \left(\frac{\Delta}{\alpha\Delta}\right)\alpha^{\alpha\Delta}(1-\alpha)^{(1-\alpha)\Delta} \stackrel{\text{Stirling}}{\approx} \frac{\sqrt{2\pi\Delta}\left(\frac{\Delta}{e}\right)^{\Delta}\alpha^{\alpha\Delta}(1-\alpha)^{(1-\alpha)\Delta}}{\sqrt{2\pi\alpha\Delta}\left(\frac{\alpha\Delta}{e}\right)^{\alpha\Delta}\sqrt{2\pi(1-\alpha)\Delta}\left(\frac{(1-\alpha)\Delta}{e}\right)^{(1-\alpha)\Delta}}$$

$$= \sqrt{\frac{1}{2\pi\alpha(1-\alpha)\Delta}} \tag{38}$$

So, even in the critical case of the P-mode, the signal is decaying, but only as a power law $O(\Delta^{-1/2})$ of the path length.

### D.3  Computational complexity analysis

For the (chunkwise) recurrent mode of operation, the state updates need to be taken into consideration. Let's ignore the S and M multiplications for the moment. For each chunk (higher level node), there are $E_{in} * E_{out}$ state transition operations (T C) and $N_{nodes}$ updates of key-value outer products. The source terms can be absorbed into K / V, and the pre-computation overhead is neglected in this analysis. This leads to a complexity per chunk for the recurrent part ($C' = \sum TC + SKV$):

$$F^{FLOP}_{rec} = N_{ein} * N_{eout} * D_K * D_V + N_{nodes} * D_K * D_V * N_{eout} \tag{39}$$

There is also the memory complexity of loaded memory parts, assuming that the loaded chunk data (K, V) can be kept in lower-level memory, iterating over edge combinations:

$$F^{MEM}_{rec} = N_{ein} * N_{eout} * D_K * D_V + N_{nodes} * (D_K + D_V) \tag{40}$$

The edge-combination memory overhead is reduced by the amount of edge/cell states that can be cut down to $(N_{ein} + N_{eout}) * D_K * D_V$. For the parallel computation ($Q \cdot C + (G \odot Q \cdot K) \cdot V$), we get the following complexities, assuming that the gating and mark terms are pre-computed with no overhead:

$$F^{FLOP}_{par} = N_{ein} * N_{nodes} * D_K * D_V + N^2_{nodes} * (D_K + D_V) \tag{41}$$

$$F^{MEM}_{par} = N_{ein} * D_K * D_V + N_{nodes}(2 * D_K + D_V) \tag{42}$$

Note that this can be further fused, if $D_K, D_V$ are small enough. Actually, for the concrete reduction of the einsums, we rely on JAX+XLA, which might peform this suboptimally. Now, if we add this up with a certain overhead $M$ per memory op (333 FLOPs/byte for a H100 on TensorCores), and add a scaling for the $N_{ein}, N_{eout} = N^\alpha_{nodes}, \alpha = 1/2$ in 2D, $N_{total} = N_{nodes} * N_{chunks}$:

$$F^{EFF} = N_{total}/N_{nodes} * [(2 * N^{1+\alpha}_{nodes} + N^{2\alpha}_{nodes} + N^2_{nodes}) * D_K * D_V \tag{43}$$

$$+ M * (3 * N^\alpha_{nodes} * D_K * D_V + N_{nodes} * (3 * D_K + 2 * D_V))] \tag{44}$$

Removing constants and scales and calculating the derivative of the leading order terms ($D_K * D_V$), we arrive at:

$$d\tilde{F}^{EFF} = 2\alpha * N^{\alpha-1}_{nodes} + (2\alpha - 1) * N^{2\alpha-2}_{nodes} + 1 + M * 3(\alpha - 1) * N^{\alpha-2}_{nodes} = 0 \tag{45}$$

which leads to an optimal number of nodes per chunk $N_{nodes}$ (derivative equals 0) for 2D ($\alpha = 1/2$):

$$d\tilde{F}^{EFF} = N_{nodes} + N^{3/2}_{nodes} - 3/2 * M = 0 \tag{46}$$

So, the optimum is:

$$(3/2 * M)^{2/3} \le N_{nodes}^{2D*} \le (3 * M)^{2/3} \tag{47}$$

For the sequential linear RNN case ($\alpha = 0$), it is:

$$N_{nodes}^{1D*} = 2\sqrt{M} \tag{48}$$

Note that lower-order terms are neglected here, and FLOPs counting might vary in prefactors, but the overall scaling of the optimum remains the same. Also, interestingly, for graphs where the edges in between chunks scale as the number of contained nodes (or worse, $\alpha \ge 1$), the derivative is always positive and the optimum is the pure recurrent computation.

Modern GPUs need a high arithmetic intensity [Williams et al., 2009, Dao et al., 2022, Pöppel et al., 2025] such that FLOPs become the bottleneck of computation. Because of this trade-off between FLOPs versus runtime + memory, typically more parallelization is beneficial for actual runtime up to a certain Pareto limit (see also Beck et al. [2025a]). For larger images, full parallelization will not be the optimal case due to the quadratic FLOPs complexity in the image size.

# E  Experimental Results

Due to the complex structure of pLSTM in 2D and its parallelization, we use a `jax`[1]-based implementation to enable efficient compilation of the computational graph. In particular, we re-use parts of Park [2024]. Early experiments on using `torch.compile` on the `torch`[2] implementation showed a slowdown rather than a speed-up of the model computations, which is why we use `jax`. Our source code, released with this work, has a detached configuration that works for both `jax` and `torch` and should enable a fast switch of frameworks for future changes. We use `jax` version `0.4.32` and CUDA `12.2`.

## E.1  Initialization

While for the ablations and arrow pointing extrapolation, we use zeros initialization for the weight terms of Source, Mark, Transition, Direct as well as Orientation (P mode angle) layers, on the DeiT-III style training on ImageNet1k, using a non-zero small normal init leads to significantly better results reported here. For ViT, we observe that LayerScale initialized with 1e-4 is important for ImageNet1k training, whereas on Arrow Pointing Extrapolation, it is important to initialize the LayerScale at 1 to reach the observed performance.

**Table 5:** General pLSTM initialization settings

| Parameter | Value |
|---|---|
| Source Bias Init | -4 |
| Mark Bias Init | -4 |
| Direct Bias Init | -6 |
| Transition Bias Init | 1 |
| Transition Scaling Factor | 5 |
| Orientation Bias Init | Headwise Range in [-2, 2] |
| Multi-Head RMSNorm $\epsilon$ | 1e-5 |
| Pooling | Corner Patches |
| Mode | P+D (alternating) |

## E.2  Arrow Pointing

Here, we train for 50 epochs with batch size 128, using learning rates [ 1e-4, 3e-4, 1e-3 ] and report the mean validation curves over five seeds at the best learning rate. For ViL, we include 1e-5 as the

---

[1] https://jax.dev
[2] https://pytorch.org

learning rate, as it fails to improve for higher learning rates. The validation datasets (standard and extrapolation) are generated from the same validation seed for all runs. We use a linear-warmup + cosine-decay schedule with one warmup epoch starting from zero and ending at $0.001 \times$ peak_lr. The models take about one hour of training on a single NVIDIA H100-64GB.

### E.2.1 Test results

**Table 6:** Test Results on Arrow Pointing Extrapolation (5 seeds, 90% confidence interval) and training time on one H100, * is just trained on one learning rate, ° is without flash attention for fair comparison

| Model | Best LR | Test Acc. | Test Acc. (Ext.) | Training Time (h) |
|---|---|---|---|---|
| pLSTM | 0.0001 | $0.972 \pm 0.003$ | $0.778 \pm 0.031$ | 2.0 |
| pLSTM / (no posemb.) | 0.0001 | $0.975 \pm 0.003$ | $0.769 \pm 0.018$ | 2.0 |
| pLSTM / (P-mode only) | 0.0003 | $0.978 \pm 0.005$ | $0.746 \pm 0.027$ | 2.0 |
| pLSTM / (D-mode only) | 0.0001 | $0.957 \pm 0.002$ | $0.828 \pm 0.028$ | 2.0 |
| pLSTM / (STM bias only) | 0.0001 | $0.975 \pm 0.003$ | $0.784 \pm 0.020$ | 2.0 |
| ViT | 0.0003 | $0.915 \pm 0.019$ | $0.707 \pm 0.014$ | $0.7^{\circ}$ |
| ViL | 3e-05 | $0.823 \pm 0.157$ | $0.503 \pm 0.004$ | 1.9 |
| EfficientNet | 0.001 | $0.998 \pm 0.001$ | $0.649 \pm 0.023$ | 1.2 |
| 2DMamba | 0.0001* | $0.964 \pm 0.045$ | $0.584 \pm 0.144$ | 8.1 |
| Mamba2D | 0.0001* | $0.553 \pm 0.121$ | $0.500 \pm 0.001$ | 7.3 |
| V2M | 0.0001* | $0.948 \pm 0.009$ | $0.557 \pm 0.036$ | 26.7 |
| V2M+Mamba2D | 0.0001* | $0.496 \pm 0.005$ | $0.500 \pm 0.001$ | 17.9 |

### E.3 ImageNet-1k

For training on ImageNet-1k, we match the original training schedule of DeiT-III [Touvron et al., 2022]. EfficientNets [Tan and Le, 2019] as a convolution-based baseline architecture are still the SOTA at these scales, but it is important to note that larger models were also trained at larger resolutions, whereas training resolution was not scaled with the models for all other models. EfficientNet, Mamba2D, and 2DMamba are non-isotropic in that their embedding dimensions are scaled up with depth. ViT, ViL, and pLSTM are isotropic, as is the reported isotropic ConvNeXt. For ConvNeXt, isotropic models showed lower performance compared to non-isotropic ones. For pLSTM, a transition to non-isotropic model could therefore lead to performance gains as well.

All of the models are pre-trained for up to 24 hours on 4 nodes, with 4 NVIDIA H100-64GB GPUs each. ViT models are faster (about 30-50%), as our models do not yet utilize specific kernels. For counting FLOPs, we use `fvcore`[3] on the PyTorch implementation. For arrow pointing extrapolation training, the Mamba2D and 2DMamba variants were about twice as slow as pLSTM despite utilizing custom kernels.

### E.3.1 Ablation Settings

For the ablation studies, we use a simplified training setting, without an additional fine-tuning stage, resembling a DeiT-T training over 400 epochs [Touvron et al., 2021]. In Table 8, we provide a summary of the used training hyperparameters.

### E.4 Graph Benchmarks

For the graph version of pLSTM, we use `torch`, as the dynamic computation graph support is better for this framework. For simplicity, we also do not implement any parallelization of the graph computation, but use the recurrent form only.

---

[3]`https://github.com/facebookresearch/fvcore`

**Table 7:** ImageNet1k - DeiT-III style training parameters

| Parameter | Value ( → Fine-Tuning) |
|---|---|
| Image Resolution | $192 \times 192 \rightarrow 224 \times 224$ |
| Training Epochs | 800 (T), 400 (S), 400 (B) → 20 |
| Hidden Dimension | 192 (T), 384 (S), 768 (B) |
| Num Heads | 3 (T), 6 (S), 12 (B) |
| DropPath Rate | 0. (T), 0.05 (S), 0.1 (B) |
| LayerScale | - |
| Warmup Epochs | 5 |
| Peak Learning Rate | 4e-3 (T), 4e-3 (S), 3e-3 (B) → 1e-5 |
| Weight Decay | 0.2 |
| Gradient Clip Norm | 1.0 |
| Optimizer | Lamb → AdamW |
| Loss Type | Binary Cross Entropy → Cross Entropy |
| MixUp | 0.8 |
| CutMix | 1.0 |
| Label Smoothing | 0.0 → 0.1 |
| Global Batch Size | 2048 → 512 |
| ColorJitter | 0.3 → 0.0 |
| AutoAugment | "3a" → "rand-m9-mstd0.5-inc1" |
| RandomErasing | 0. |
| AugmentRepeats | 3 → 1 |
| TestCropRatio | 1.0 |
| RandomCrop | rrc |

**Datasets.** Our experiments are conducted with 10-fold cross-validation on popular TUDatasets [Morris et al., 2020]. For each fold, we use $1/10$ of the respective training data as test set, $1/10$ for validation and $8/10$ for training. For every fold, we pick the test accuracy of the epoch with the best validation accuracy and report the average over all folds. For pLSTM, we encode the number of neighbors for each node with a standard positional encoding [Vaswani et al., 2017]. Otherwise, no data transformation, augmentation, or normalization is used.

**Training procedure.** We train all models for 100 epochs with AdamW, a learning rate of 0.001, batch size of 64, and a cosine decay learning rate schedule with 5 warmup epochs.

**Models.** The compared models are GAT [Veličković et al., 2018], GCN [Kipf and Welling, 2017], GIN [Xu et al., 2019], LSTM GNN [Liang et al., 2016], MPNN [Gilmer et al., 2017]. All GNNs consist of an encoder, decoder, and 4 or 8 message passing layers. The best model configuration is selected based on the validation accuracy. pLSTM also consists of the same encoder and decoder, but has 2 layers that operate in D-mode and 2 layers that operate in P-mode. To achieve an approximately similar parameter count, we fix the hidden dimension of pLSTM to 96, while the other GNNs have a hidden dimension of 128. All trained models have a parameter count of < 300,000.

**Graph Positional Embedding.** Encodings are frequently used to boost performance on graph benchmarks. For example, an encoding based on eigenvectors and eigenvalues of the graph Laplacian (LapPE) [Dwivedi and Bresson, 2020] has been frequently used to improve results on peptides-struct, while a random-walk structural encoding (RWSE) performs well on peptides-func [Dwivedi et al., 2021, Rampášek et al., 2022]. Especially the RWSE could skew the results, as it is very similar to message passing (which we compare against). However, in our experiments, we do not include LapPE or RWSE for pLSTM or any other method for a cleaner architectural comparison. In this setting, pLSTM performs comparable or better than several baselines on the peptide benchmarks. pLSTM is also fully compatible with such encodings, and integrating them could further improve performance. However, we intentionally did not include them in this study to better isolate the effect of our proposed architecture incorporating the structural information.

**Table 8:** ImageNet1k - DeiT style ablation training parameters

| Parameter | Value |
| --- | --- |
| Image Resolution | 224 × 224 |
| Training Epochs | 400 |
| Hidden Dimension | 192 |
| Num Heads | 3 |
| DropPath Rate | 0. |
| LayerScale | - |
| Warmup Epochs | 5 |
| Peak Learning Rate | 1e-3 |
| Weight Decay | 0.05 |
| Gradient Clip Norm | 1.0 |
| Optimizer | AdamW |
| Loss Type | Cross Entropy |
| MixUp | 0.8 |
| CutMix | 1.0 |
| Label Smoothing | 0.0 → 0.1 |
| Global Batch Size | 2048 |
| ColorJitter | 0.0 |
| AutoAugment | "rand-m9-mstd0.5-inc1" |
| RandomErasing | 0.25 |
| AugmentRepeats | 3 → 1 |
| TestCropRatio | 1.0 |
| RandomCrop | rrc |

