# OpenReview forum: "pLSTM: parallelizable Linear Source Transition Mark networks"
_NeurIPS.cc/2025/Conference — NeurIPS 2025 poster_

### Official Review · Reviewer_hu73 · 2025-07-01

**Clarity:** 3
**Significance:** 3
**Originality:** 3
**Rating:** 4
**Confidence:** 3

**Summary:**

This paper demonstrates a gated linear recurrent network with associative cell, similar to MAMBA, designed for arbitrary DAGs.  Two modes of propagation are proposed: a D-mode that is similar to 2DMAMBA and has the same decay properties assuming the transition matrix is norm-constrained as described there, and a P-mode that propagates with decay exponential in the path length in an arbitrary learned direction provided that the L1-norm of each column of the transition matrix is less than 1.  Results are demonstrated for three applications: On ImageNet-1K it performs within 4% of SoTA, on several small-molecule graph sets it performs within 1% of SoTA, and on a new long-distance 2D inference task it outperforms ViT.

**Questions:**

Does 2DMAMBA also outperform pLSTM for the arrow-pointing task?

**Ethical Concerns:**

["NO or VERY MINOR ethics concerns only"]

**Limitations:**

Yes

**Quality:**

3

**Strengths And Weaknesses:**

Strengths:

The framework elegantly generalizes linear gated recurrent networks to arbitrary graphs. The proof of non-explosion is clear and elegant and might even imply a statement more general than what is claimed here.

Weaknesses:

There is a lot of recent research trying to generalize linear gated recurrent units to 2D.  Although this formalism is more elegant than others I've seen, its performance considerably lags 2DMAMBA.

Minor issues/ possible typos:

The second term on RHS in both Eq. (6) and Eq. (7) has no factor with a "v" subscript, suggesting that this term is constant w.r.t. the v subscript on the LHS.  That seems wasteful and inconsistent.  The recurrent terms suggest that V_k^n should have been V_v^n, whereas the phrase "skip directly from input to output" suggests that Q_kK_kV_k should have been Q_vK_vV_v; I wonder if this is a typo, and if so, which correction is correct?

Is Eq. (8) intended to be approximate?  It is true if |s| means any vector norm and |T| means the consistent matrix norm, but it is untrue for some other common interpretations of those symbols, e.g., I have often seen |T| used to denote the matrix determinant, which would make Eq. (8) untrue. It seems that Eq. (8) is only really used to define the P-mode in Eq. (9), since the D-mode uses a different derivation, but if you were to use double-bars instead of single-bars in Eq. (8), it could be interpreted as correctly licensing other propagation modes besides P-mode and D-mode (specifically, it seems to license any propagation mode in which ||T|| is the matrix norm consistent with ||s|| and ||T||<=1).

---

> ### Author Rebuttal · Authors · 2025-07-31
>
> Thank you for your review and, in particular, the detailed look at the presented equations.
>
> Indeed, in the RHS terms of Eq. (6) and (7), the $V$ term should have subscript $V_{v}$ instead of $V_{k}$. Thank you for pointing out this typo. We have now fixed this in our updated version of our manuscript.
>
> ## P-mode norm choice and notation
>
> Regarding the norms in Eq. 8, these are intended to denote any vector norm and a respective consistent matrix norm (we have updated to a double | notation). You are right that this allows for multiple options for the P-mode. Note, however, that there are only two matrix norms that can be computed locally without iterating over the graph globally: The $L_{1}$ and $L_{\inf}$ norms. Note that these are also dual to each other, which provides both stability for the activations and gradients (see l.218ff). One could flip the norms for activations and gradients, leading to a slightly different P-mode indeed. But, for other norms, the global graph structure would need to be considered, which creates an undesirable additional overhead. Therefore, we restrict the norm to $L_{1}$ for our proposed P-mode.
>
> ## Additional arrow pointing extrapolation results
>
> Thank you for the question. We address 2DMamba's performance on the arrow-pointing task in our response to Reviewer T3S3, where we include additional results. Notably, 2DMamba underperforms pLSTM on this task, despite its stronger results on ImageNet.
>
> ## ImageNet improvements
>
> We also improved pLSTM's ImageNet results with modified initialization (see T3S3 response). Importantly, pLSTM is trained in an isotropic setting (no hidden dimension upscaling), whereas 2DMamba, V2M, and Mamba2D use non-isotropic configurations, which are known to boost performance (e.g., in the V2M paper). In the revised version, we will clearly mark this distinction.
>
> Thank you again for your positive assessment of our work. If any questions remain or come up during the discussion period, we would be happy to engage in further discussion.

---

> ### Author Response · Authors · 2025-08-07
>
> As the discussion period draws to a close, we wanted to sincerely thank you for your time and thoughtful feedback. We've done our best to address your concerns in the rebuttal and hope that our responses have clarified the key points of the paper.
> If there are any remaining questions or uncertainties, we would be happy to elaborate further while the discussion is still open. Should you feel that your concerns have been addressed, we would greatly appreciate your consideration in reflecting that in your final evaluation.
> Thank you again for participating in the review process. We truly value your input.

---

### Official Review · Reviewer_T3S3 · 2025-07-01

**Clarity:** 4
**Significance:** 2
**Originality:** 3
**Rating:** 4
**Confidence:** 2

**Summary:**

The authors propose an innovative architecture that extends linear RNNs to Directed Acyclic Graphs (DAGs), enabling chunkwise-parallel processing. To address parallelization on DAGs, they introduce two distinct parallelization strategies: the Propagation Model (P Model), which covers all paths but faces efficiency constraints, and the Distribution Model (D Model), which retains only a subset of paths for enhanced parallelism. Experimental results on both 2D (image) and graph data demonstrate that this architecture surpasses Vision Transformers (ViT) and achieves performance comparable to popular state-of-the-art models.

**Questions:**

See above.

1. Please explain about the efficiency of the architecture. What is the theoretical computational complexity for each parallelization policy? Why the experiments do not show the better efficiency?
2. More results on arrow datasets and graph datasets.

**Ethical Concerns:**

["NO or VERY MINOR ethics concerns only"]

**Final Justification:**

Because my main concern has been resolved, I will raise my score to 4.

**Limitations:**

yes

**Quality:**

3

**Strengths And Weaknesses:**

Strength:
1. The architecture is novel. The idea to extend the linear RNN to non-regular DAGs data is interesting. By combining the two distinct model (P model and D model), different propagation paradigms can be constructed for structured data.
2. The architecture promise well parallelizablity.

Weakness:
1. The most weakness is the not outstanding experiment results for both performance and the efficiency. The motivation of linear RNN is the efficiency. But the flops results shown in the experiments show similar efficiency compared to other models. And the experiments on graph data lacks the discussion about the efficiency.
2. In the arrow pointing experiments, the only one baseline is the ViT, why other baseline models are not tested on this datasets?
3. The architecture is proposed for DAGs, but the main experiments are on image datasets. I think more experiments on graph datasets could help the authors to proof their novelty and significance. At the same time, both the accuracy and the efficiency should be discussed.

---

> ### Author Rebuttal · Authors · 2025-07-31
>
> We thank the reviewer for the constructive feedback. We are glad that you consider our architecture novel and the extension to non-regular DAGs interesting.
>
> We have now extended the arrow-pointing extrapolation experiments to include additional baselines. As shown, pLSTM outperforms all other baselines by a clear margin on the extrapolation task, while remaining competitive in terms of training time.
>
> | Model          |  Test acc. (ext) - 5 runs | Training time (h) |
> |----------------|---------------------------|-------------------|
> | EfficientNet   |  0.649 ± 0.023            |       1.2         |
> | ViT            |   0.707 ± 0.014           |       0.4         |
> | ViL            |  0.503 ± 0.004            |       1.9         |
> | Mamba2D        |  0.500 ± 0.001            |       7.3         |
> | 2DMamba        |  0.584 ± 0.144            |       8.1         |
> | V2M            | 0.557 ± 0.036             |      26.7         |
> | V2M + Mamba2D  |  0.500 ± 0.001            |      17.9         |
> | pLSTM          |   **0.778 ± 0.031**       |       2.0         |
>
> Moreover, we are happy to provide additional results on ‘Peptides-struct’ and ‘Peptides-func’ as new graph datasets part of the Long Range Graph Benchmark:
>
> | Model                |     Peptides-struct (MAE)↓    |     Peptides-func (Avg. Prec.)↑   |
> |----------------------|-------------------------------|-----------------------------------|
> | GAT                          |  0.270       |  0.525     |
> | GCN                         |  0.264     |  0.534    |
> | GIN                          |  0.328        |  0.593   |
> | LSTM-GNN             |   0.274                                        |  0.502 |
> | MPNN                 |   0.260                                     |  0.557    |
> | pLSTM              |  0.264                                      |  0.536  |
>
> pLSTM can achieve results similar to re-trained baseline models on these benchmarks, being the second best for the ‘Peptides-struct’. We encoded atoms as nodes and bonds as edges and did not use any additional encoding such as Laplacian encoding or RWSE. This has a negative effect on the results, however it allows for a more precise comparison without any confounding factors.
>
>
> Additionally, we observed that a minor change to the weight initialization (increasing the initialization scale) improves pLSTM's performance on ImageNet-1k, particularly at larger model sizes. This change likely introduces a beneficial regularization effect via increased activation diversity.
> We would like to emphasize that our primary goal is not to outperform current state-of-the-art architectures, but rather to introduce a novel and generalizable architectural mechanism. Our work opens the door for future improvements and integration into broader model designs. Note also that both Mamba2D and 2DMamba use non-isotropic backbones, which often yield stronger performance at scale, making the performance of isotropic pLSTM even more encouraging.
>
> | Model      | Test acc. |
> |-------------|-------------|
> | pLSTM-T |      75.2  |
> | pLSTM-S |       80.7 |
> | pLSTM-B |       82.5 |
>
> ## Efficiency considerations
> Thank you for the question. The parallelization scheme for DAGs was not used in our graph benchmark experiments due to the irregularity of the graph structures. Instead, we used a recurrent implementation, which has low FLOP complexity but higher runtime due to limited parallelism.
>
> For structured data like 1D/2D grids, we do implement efficient parallel versions, which benefit from reduced memory overhead and fast batched operations. We believe the parallelization scheme has strong theoretical potential and plan to explore efficient implementations for general graphs in future work.
>
> ## FLOPs considerations
> For the (chunkwise) recurrent mode of operation, the state updates need to be taken into consideration. Let's ignore the S and M multiplications for the moment.
> For each chunk (higher level node), there are $E_{in} * E_{out}$ state transition operations $(T * C)$ and $N_{nodes}$ updates of key-value outer products. The source terms can be absorbed into K / V, and the pre-computation overhead is neglected in this analysis. This leads to a complexity per chunk for the recurrent part ( $C' = \sum T \cdot C + S K V$ ):
>
> $ F_{rec}^{FLOP} = N_{ein} * N_{eout} * D_{K} * D_{V} + N_{nodes} * D_{K} * D_{V} * N_{eout}$
>
> There is also the memory complexity of loaded memory parts, assuming that the loaded chunk data (K, V) can be kept in lower-level memory, iterating over edge combinations:
>
> $F_{rec}^{MEM} = N_{ein} * N_{eout} * D_{K} * D_{V} + N_{nodes} * (D_{K} + D_{V})$
>
> The edge-combination memory overhead is reduced by the amount of edge/cell states that can be cut down to $(N_{ein} + N_{eout}) * D_{K} *D_{V}$ .
> For the parallel computation ($Q \cdot C + (G \odot Q \cdot K) \cdot V$ ), we get the following complexities, assuming that the gating and mark terms are pre-computed with no overhead, and all :
>
> $F_{par}^{FLOP} = N_{ein} * N_{nodes} * D_{K} * D_{V} + N_{nodes}^2 * (D_{K} + D_{V})$
>
> $F_{par}^{MEM} = N_{ein} * D_{K} * D_{V} + N_{nodes} (2 * D_{K}  + D_{V})$
>
> Note that this can be further fused if $D_{K}, D_{V}$ are small enough. Actually, for the concrete reduction of the einsums, we rely on JAX+XLA, which might perform this suboptimally.
> Now, if we add this up with a certain overhead $M$ per memory op (333 FLOPs/byte for a H100 on TensorCores), and add a scaling for the $N_{ein}, N_{eout} = N_{nodes}^{\alpha} , \alpha = 1/2$ in 2D, $N_{total} = N_{nodes} * N_{chunks}$:
>
>
> $F^{EFF} = N_{total} / N_{nodes} * ( (2 * N_{nodes}^{1+\alpha} + N_{nodes}^{2 \alpha} + N_{nodes}^{2}) * D_{K} * D_{V}  +  M * (3 * N_{nodes}^\alpha * D_{K} * D_{V}   + N_{nodes} * ( 3 * D_{K} + 2 * D_{V} )))  $
>
> Removing constants and scales and calculating the derivative of the leading order terms (D_K * D_V), we arrive at:
>
> $d\tilde{F}^{EFF} = 2 \alpha * N_{nodes}^{\alpha-1} + (2 \alpha - 1) * N_{nodes}^{2 \alpha - 2} + 1  +  M * 3 (\alpha - 1) * N_{nodes}^{\alpha - 2} = 0$
>
> which leads to an optimal number of nodes per chunk $N_{nodes}$ (derivative equals 0) for 2D ($\alpha = 1/2$):
>
> $d\tilde{F}^{EFF} = N_{nodes} + N_{nodes}^{3/2}  -  3/2 * M = 0$
>
> So the optimum is:
>
> $(3/2 * M )^{2/3} \leq N_{nodes}^{2D*} \leq (3 * M)^{2/3}$
>
> For the sequential linear RNN case ($\alpha = 0$), it is:
>
> $N^{1D*}_{nodes} = 2 \sqrt{M}$
>
> Note that lower-order terms are neglected here, and FLOPs counting might vary in prefactors, but the overall scaling of the optimum remains the same.
> Also, interestingly, for graphs where edges in between chunks scale as the number of contained nodes (or worse, $\alpha \geq 1$), the derivative is always positive and the optimum is the pure recurrent computation.
>
> Modern GPUs need a high arithmetic intensity such that FLOPs become the bottleneck of computation. Because of this tradeoff between FLOPs vs. runtime+memory, typically more parallelization is beneficial for actual runtime up to a certain Pareto limit (see [TFLA]). For larger images, full parallelization will not be the optimal case due to the quadratic FLOPs complexity in the image size.
> Thank you for raising this important point. We have added an extended version of this analysis to our updated manuscript.
> Note that we integrated several other models now into our codebase, and the training time for the arrow pointing experiments shows that this is not strongly correlated with FLOPs (B=128, R=192x192, 1 NVIDIA H100 GPU), but rather implementation efficiency.
>
> Our manuscript has improved considerably by incorporating your feedback: thank you. If you find our responses and additional results useful, we would appreciate it if you would consider changing your score.

---

> > ### Comment · Reviewer_T3S3 · 2025-08-04
> >
> > I am not an expert about the model architecture so I cannot determine whether the architecture is promising. But for me, the current experiment results is not satisfying. In my opinion, even the ViT is not the state-of-the-art model and your method surpasses ViT with 7% accuracy but 4 times training time. And on graph dataset, it fall short in both performance and efficiency compared to even a naive MPNN. So I will keep my score as 3. But I have to say I cannot determine whether it is potentially useful so if other reviewers believe the performance shows that it is promising in the future, I am OK that it is accepted.

---

> > > ### Author Response · Authors · 2025-08-04
> > >
> > > Thank you for your honest feedback and for carefully considering the trade-offs. We understand your concerns regarding performance and training time.
> > > We would like to clarify a few points:
> > > 1) The training time for pLSTM in the arrow-pointing task is $\sim$2 hours on a single H100 GPU. While it is longer than ViT ($\sim$0.4h), it remains significantly faster than most other strong baselines (e.g., V2M: 26.7h, 2DMamba: 8.1h, Mamba2D: 7.3h), despite not using custom kernels (like FlashAttention). Our goal was to highlight generalization, not just peak performance, on a challenging extrapolation task where ViT and others underperform.
> > > 2) On graph datasets, pLSTM achieves comparable results to re-trained baselines without additional encodings (e.g., Laplacian, RWSE), which are commonly used to boost performance. This design choice was intentional to isolate architectural impact.
> > >
> > > More importantly, pLSTM introduces a novel parallelization framework for linear networks on general DAGs, enabling new directions in multi-dimensional recurrent modeling leveraging the graph structure. While performance is not yet superior in all cases, we believe our work lays the foundation for future advancements, similar to how early Transformers were initially slower and underperformed LSTMs on certain tasks.

---

> > > > ### Comment · Reviewer_T3S3 · 2025-08-05
> > > >
> > > > I agree that a new architecture will meet many problem like no custom kernel acceleration and not very suitable infra. I think maybe one way to make fair comparison could be do not use these acceleration like FlashAttention.
> > > >
> > > > As for graph datasets, do you mean the MPNN also benefit from these additional encoding? So can you explain what is these additional encoding and can these baseline run without these additional encoding? Or can your methods also use these additional information?

---

> > > > > ### Author Response · Authors · 2025-08-06
> > > > >
> > > > > Thank you again for your thoughtful follow-up and for raising important points.
> > > > > 1) On runtime fairness:
> > > > > As requested, we have rerun the ViT baseline __without using FlashAttention__ (i.e., with standard PyTorch matmuls and softmax). This increases its training time from 0.4h to 0.7h on the arrow-pointing task. In comparison, __pLSTM completes training in 2.0h__, which is still significantly faster than other strong baselines (e.g., V2M: 26.7h, Mamba2D: 8.1h), while delivering stronger generalization performance on the extrapolation task. ViT performs competitively at small scales due to efficient matrix multiplications, but scales quadratically, unlike pLSTM’s linear scaling (as discussed in our theoretical analysis), making pLSTM better suited for larger images or structured data.
> > > > >
> > > > > 2) On graph datasets and encodings:
> > > > > Encodings are frequently used to boost performance on graph benchmarks. For example, an encoding based on eigenvectors and eigenvalues of the graph Laplacian (LapPE) has been frequently used to improve results on peptides-struct, while a random-walk structural encoding (RWSE) performs well on peptides-func. Especially the RWSE could skew the results, as it is very similar to message passing (which we compare against). However, in our experiments, we do __not include LapPE or RWSE for pLSTM or any other method__ for a cleaner architectural comparison. In this setting, pLSTM performs comparably or better than several baselines on the peptide benchmarks.
> > > > > pLSTM is also fully compatible with such encodings, and integrating them could further improve performance. However, we intentionally did not include them in this study to better isolate the effect of our proposed architecture incorporating the structural information.
> > > > >
> > > > > We hope this clarifies the tradeoffs and motivations behind our experimental setup. While we acknowledge that pLSTM may not yet outperform all baselines in every scenario, we believe it offers a novel and generalizable mechanism that opens new directions for modeling on structures beyond (and including) sequences. As with early architectures, we expect performance to improve further with engineering and optimization. We sincerely appreciate your engagement and constructive feedback throughout the discussion.

---

> > > > > > ### Comment · Reviewer_T3S3 · 2025-08-06
> > > > > >
> > > > > > Can you give me a short explanation to support your claim that "pLSTM is also fully compatible with such encodings, and integrating them could further improve performance"? If so, I will raise my score. Or if other reviewers or AC believe it is correct and obvious, I think it is OK.

---

> ### Author Response · Authors · 2025-08-06
>
> Thank you for your thoughtful follow-up. To support our claim:
>
> Yes, pLSTM is fully compatible with standard structural encodings, such as Laplacian positional encodings (LapPE) [1] and Random Walk Structural Encodings (RWSE) [2, 3].
>
> 1) For LapPE and basic RWSE, the node features are enhanced by precomputed positional encodings (e.g., Laplacian eigenvectors or random walk probabilities), which are simply added to or concatenated with the input embeddings. This is directly compatible with pLSTM and requires no architectural changes.
>
> 2) For more expressive RWSE variants [3], which introduce additional edge features or virtual edges (e.g., between distant nodes in the same structure), pLSTM can naturally operate on the extended graph by incorporating these edges and features in its source and transition computations. This is because the gating functions in pLSTM are flexible and can integrate arbitrary edge features.
>
> In summary, incorporating these encodings into pLSTM is straightforward and opens up clear directions for improving performance. We did not include them in our current experiments to isolate the architectural contribution of pLSTM itself, but we agree that adding such encodings would likely lead to stronger results.
>
> Thank you again for the engaging discussion and your openness to revising the score.
>
> [1] A Generalization of Transformer Networks to Graphs, Dwivedi et al., 2021
>
> [2] Graph Neural Networks with Learnable Structural and Positional Representations, Dwivedi et al., 2022
>
> [3] Recipe for a General, Powerful, Scalable Graph Transformer, Rampášek et al., 2022

---

> > ### Author Response · Authors · 2025-08-07
> >
> > As the discussion period draws to a close, we wanted to sincerely thank you for your time and thoughtful feedback. We've done our best to address your concerns in the rebuttal and hope that our responses have clarified the key points of the paper.
> > If there are any remaining questions or uncertainties, we would be happy to elaborate further while the discussion is still open. Should you feel that your concerns have been addressed, we would greatly appreciate your consideration in reflecting that in your final evaluation.
> > Thank you again for participating in the review process. We truly value your input.

---

### Official Review · Reviewer_hoGY · 2025-07-03

**Clarity:** 4
**Significance:** 4
**Originality:** 3
**Rating:** 5
**Confidence:** 4

**Summary:**

This paper extends recent sequence models based on linear RNNs and linear attention to graph structured data, particularly directed acyclic graphs (DAGs). In the introduced pLSTM framework hidden states are ascribed to edges and propagate through nodes depending on source (=input), transition (=forget), and mark (=output) gates. The linearity of the transition function allows to unroll this recurrent description (under topological ordering) into the quadratic form of linear attention with a connectivity-dependent decay/gating term. This enables both an analysis of potential parallelization and two vanishing/exploding signal stabilization techniques (P-Mode and D-Mode). Finally, the framework is instantiated and evaluated on images (2d-regular grid) and molecular graph tasks.

**Questions:**

- Why is it important to associate the states with edges instead of the nodes? From what I understand, the source and mark gates could also be subsumed into KV and Q respectively without significantly impacting Equations 5,6,7. This would also avoid extra notation on line-graphs in Sec. 4.2.
- Would your analysis based on power iterations allow for other (potentially input-dependent) stabilization techniques beyond P-mode and D-mode? What would be the effect of limiting the norm of the row (i.e. node local input normalization)? Would row-wise softmax recover GATs? Would the exponential gating with output normalization of xLSTM also be stable?
- Could you compare V2M and Mamba2D to pLSTM on the arrow task? Since the final model interleaves D-Mode and P-Mode layer, could you furthermore compare to a model with interleaved V2M and Mamba2D layers?
- In recent gated linear attention models [2,3], an important aspect for parallelization is that the gating matrix $G^{nn’}$ is a rank-1 structure which allows to decompose an outer product and subsume the resulting vectors into the key and query terms. Significant speedups are realized by using the associative scan only once to precompute cumulative gating as a preprocessing step. Can you explain how this translates to parallelization on DAGs and how you can compute pLSTM on regular grids (in particular the 1D case) without cumulative sums/products?
- Can you explain how your parallelization scheme differs from applying parallel kernels along each dimension independently, as is (presumably) done in the works that you discussed as most similar to yours (2DMamba, V2M, Mamba2D)?

------------

[2] Yang et al. (2023), *“Gated Linear Attention Transformers with Hardware-Efficient Training”*, https://arxiv.org/abs/2312.06635

[3] Dao et al. (2024), *“Transformers are SSMs: Generalized Models and Efficient Algorithms Through Structured State Space Duality”*, https://arxiv.org/abs/2405.21060

**Ethical Concerns:**

["NO or VERY MINOR ethics concerns only"]

**Final Justification:**

The authors addressed my main concerns: potential extensions of pLSTM for other stabilization modes and how their parallelization method relates to previous works.

Overall I believe that the pLSTM graph mixing framework with its novel stabilization analysis and parallelization technique poses a significant and unique contribution for the field. Although the evaluation is still limited, I believe that the results are promising especially since confounding factors such as backbone and hyperparameters generally favour established architectures over emerging ones. I would, however, recommend to the authors to further shift the focus from introducing a new architecture (which evokes an expectation for sota results on benchmarks) to introducing a novel framework for stabilization/parallelization (where promising results on benchmarks suffice). Since the authors addressed most of my concerns regarding the framework perspective, I increase the score to 5.

**Limitations:**

Beyond the limitations already discussed in the paper, a discussion of the potential for parallelization in practice would be interesting. This is particularly important, as the parallel algorithm for general Graphs is purely conceptual without practical evidence that speedups could be realized as opposed to the recurrent implementation.

**Paper Formatting Concerns:**

No formatting concerns.

**Quality:**

3

**Strengths And Weaknesses:**

**Strengths**
- The pLSTM framework provides a precise yet intuitive generalization of linear RNNs and linear attention to DAGs. It particularly avoids unnecessary complexities introduced by state-space model (SSM) perspective such as discretization.
- Stability guarantees in forward and backward pass can then be derived through general conditions on the (input-dependent) values of the adjacency matrix. This understanding potentially enables design and analysis of various stabilization techniques beyond the parametrizations induced by SSM discretization.
- Promising results for two instantiations of the framework in very distinct application domains highlight the potential of the approach.

**Weaknesses**
- A comparison to e.g. Mamba for graph data is missing (cf. survey [1]) and on 2D grids, the resulting stabilization techniques (P-mode and D-mode) turn out to be very minor modifications to existing works, as depicted in Figure 1. Therefore, for the resulting pLSTM models, it could be better discussed what the key novelties are. **Update:** *addressed in the rebuttal*
- Along that line, the empirical evaluation does not provide a complete understanding how pLSTM models compare to related work. In particular, the baselines for Graph data are all more than 7 years old. On the 2D arrow task which tests spatial reasoning, the claimed conceptual advantage over works which are considered most related (2DMamba, V2M, Mamba2D) is not empirically verified. **Update:** *partially addressed in the rebuttal*

**Remarks**
- In Figure 3, it would be helpful to include a green triangle for the edge on the left and variable names $n’$, $e$, $n$, $z\_n$, $h\_n$, and $e’$, $n$, $e$, $K^nV^n$, $H^n$ to the graph elements to support understanding of Equation 1,2 and 5,6. The caption could be more descriptive.
- There is an inconsistency in the notation of $T^n\_{ee’}$. While Equation 5 uses $T\_{\\text{outgoing, incoming}}^{\\text{node}}$, Equation 7 uses (mistakingly?) $T\_{\\text{incoming, outgoing}}^{\\text{node}}$. That Sec. 4.2 switches the convention to using $e’$ as outgoing and $e$ as incoming edge complicates things further. I would suggest to maintain the original notation $T^n\_{ee’}$ and explicitly mention *“from edge $e’$ to $e$”* in L200. In L213,  $T^n\_{e’e}$ is (mistakenly?) used as the adjacency matrix $\mathbf{T}$.
- From Equation 5,6 to 7, the order of symbols is reversed which makes it harder to follow and possibly caused notational inconsistency above. Maintaining the original QKV form and assuming topological ordering would make it more clear that $G^{nn’}$ is a graph-causal gating matrix which is applied element-wise to the attention matrix QK.
- The conceptual step from DAGs to (bidirectional) grids or general graphs is not very well explained. In particular the notation switches completely from $T\_{\\text{edge}}^{\\text{node}}$ to a different grid 2D specific notation with higher-level tensors which is only introduced in the appendix. Finally it is unclear how a DAG pLSTM layer is applied to general graphs.
- State Tracking Extension seems very tangential. In theory the diagonal transition of any linear RNN can always be trivially replaced by a (structured) dense transition matrix. In practice however, the difficulty lies in efficient parallelization long-range stability.

**Conclusion**
I believe that the concepts introduced in this paper will have a positive impact and therefore recommend it for acceptance. I would, however,  recommend to the authors to focus more on how their framework can contribute to novel stabilization and parallelisation schemes.

----------

[1] Atitallah et al. (2024), *“Exploring Graph Mamba: A Comprehensive Survey on State-Space Models for Graph Learning”*, https://arxiv.org/abs/2412.18322

---

> ### Author Rebuttal · Authors · 2025-07-31
>
> Thank you for your detailed feedback and your questions.
>
> ## Notation inconsistencies and visual improvement
> We have integrated your helpful feedback on the descriptive Figure 3, and the inconsistency of the T matrix notation is fixed now, where of in-coming and out-going edge index was switched. Also we added the suggested notation phrase “from in-coming edge e’ to out-going edge e”. We also made the order of S, Q, K, V consistent across all equations.
>
> ## DAGs to pLSTM
> Regarding the transition from a description of general DAGs to pLSTM on the 2D grids, we agree that the hierarchy used for parallelization could be better introduced in the grid case. The reference to Figure 5 should be emphasized as well, as the visualization greatly helps the understanding of the parallelization hierarchy. We have now revise our manuscript to improve the introduction of the grid case, moving parts of the additional notation to the appendix as it is not needed for understanding in the main paper, only for the full formulae in the appendix.
>
> ## State-tracking extension
> The state tracking extension is indeed tangential to the main contributions and is added for clarity on how to extend this framework further for DeltaNet and other architectures. Note that additional gating is crucial here (in P-mode), such that activations remain stable.
>
> ## pLSTM vs. Mamba-based models
> We have added a discussion of Mamba and Transformers applied to graph data for a broader conceptual comparison. Note that other works apply Mamba on graph data merely via randomly chosen paths in the graph. Regarding the 2D grid case (images), we acknowledge and cite the previous works on extending Mamba to 2D. pLSTM provides a unifying framework for these approaches and generalizes the applicability to DAGs. Also, pLSTM introduces a general principle of parallelization not explored in these works, as well as an extended understanding of stabilization of linear networks on DAGs (and other grids), including a geometrical interpretation thereof.
>
> Regarding additional experiments on graph data, we have results on Peptides-struct and Peptides-func. Also, on these larger-scale benchmarks, pLSTM performs well, despite no use of structural embeddings/encodings. For the results, see the answer to reviewer T3S3.
>
> ## Association of recurrent states with edges
>
> Associating states with edges rather than nodes is indeed a deliberate design choice, primarily motivated by the goal of enabling efficient parallelization and directional information flow.
>
> While traditional recurrent models (e.g., MPNNs) often associate states with nodes, this becomes limiting when trying to model directional transitions across complex structures like DAGs or grids in a parallelizable way. In our framework, information is propagated between larger regions. By tying states to edges, we naturally capture the directionality of transitions, crucial for enabling our directed propagation mode (P-mode). Without associating state with edges, transitions would either be direction-agnostic or require duplicating logic for each edge type (e.g., incoming vs. outgoing), leading to inefficiencies and loss of structure-specific inductive bias.
>
> Regarding the Source and Mark gates: while it is possible to fold them into standard KV/Q formulations, we chose to represent them explicitly for structural clarity and interpretability. Together with the Transition scalar, they form a complete gating mechanism and offer points of fast adaptation similar to learnable residual pathways. This decomposition helps separate concerns (source selection, direction marking, and propagation strength), making the model easier to analyze and potentially more flexible.
>
> ## Other stabilization modes
> Thank you for the insightful questions. Yes, our power iteration-based analysis provides a flexible framework that can accommodate various stabilization techniques beyond the proposed P-mode and D-mode.
>
> Using row-wise normalization, as you suggest, would correspond to controlling the $ L_{\inf} $ norm (affecting activations) instead of the $ L_{1} $ norm (affecting gradients), offering an alternative tradeoff in stability. This opens the door to input-dependent normalizations such as softmax, which would resemble GAT-like mechanisms. However, GAT also includes self-loops and nonlinearities, which are not present in our formulation. Our design omits these explicitly to preserve long-range propagation. #CHANGE#
> Similarly, xLSTM-style exponential gating with output normalization is compatible with our formulation. Such normalization can be applied externally to the pLSTM core, for instance by adapting the input gates accordingly. Normalizer terms can be computed as running pLSTM with $ V_v = 1 $, stabilization of an exponential Source gate (<-> input gate) here can be achieved by subtracting the maximal pre-activation value from all others. This allows pLSTM to inherit desirable stability properties while maintaining modularity.
>
> In general, the modular structure of pLSTM supports such augmentations and we believe further exploration of these stabilization variants is a promising direction for future work.
>
> ## Arrow pointing results
> In the response to reviewer T3S3, we include additional results on arrow pointing extrapolation for other models: Mamba2D, 2DMamba, V2M, ViL, V2M+Mamba2D, and EfficientNet. pLSTM outperforms all of these on the extrapolation test set due to its superior inductive bias on directional long-range interaction. We also found that a small update on weight initialization strongly improves ImageNet-1k training results for the larger scales, now outperforming the ViL baseline.
>
> ## Structural difference between the difference scans / parallelization forms
> Our parallelization strategy differs significantly from those used in 2DMamba, V2M, and Mamba2D, both in efficiency and in how directional dependencies are handled.
> V2M performs two sequential 1D scans (along rows and columns), often followed by naive parallel fusion. This approach is inherently limited by the sequential nature of 1D scans and requires explicit materialization of many intermediate states, which leads to increased memory usage and reduced arithmetic intensity, suboptimal for GPU execution.
> Mamba2D adopts a wavefront (diagonal) parallelization strategy. While this reduces the depth of sequential steps from O(P) to O(sqrt(P)), it still scales poorly with larger input sizes and lacks full parallelism, particularly when compared to scan-based or logarithmic-time alternatives.
> 2DMamba is structurally similar to V2M, but fuses some operations in 2D according to the GPU structure, which might still be sub-optimal for linear RNNs regarding arithmetic intensity.
>
> In contrast, pLSTM generalizes associative scan-style parallelism to DAGs and 2D grids by operating on the line graph representation. For regular grids (2D), this enables precomputing the gating matrix in O(½ log P) sequential steps, using tensorized operations like einsum, concatenation and padding.
> The gated/weighted aggregation step, analogous to masked linear attention, is performed in a single pass with O(P^2) FLOPs. In practice, we can use chunkwise-recurrent formulations to reduce the FLOP complexity to O(P) complexity for large images.
> Although chunking was not used in our experiments (up to 384×384 images), we instead performed a full fusion over all four DAG directions (right-bottom, left-bottom, right-top, left-top), allowing us to consolidate four fully parallel traversals into one unified computation with complete gating precomputation.
>
>
> ## Graph Parallelization Limitation
> Thank you for pointing this out. We agree that the parallel algorithm for general DAGs remains conceptual at this stage and that practical implementation and benchmarking are important directions for future work.
>
> That said, we emphasize that our current recurrent implementation for general DAGs is significantly slower than the fully parallelized baselines (e.g., CNNs, ViTs, Mamba2D). This clear performance gap strongly suggests that substantial speedups are achievable once parallelization is realized. Moreover, our method is structurally designed to enable parallel execution using line-graph representations and associative scan-style operations—tools already optimized in many modern compute frameworks.
>
>
> ## Creation of the DAG from a regular graph
> A general undirected graph $G$ can be converted into a directed acyclic graph $\mathrm{DAG}$ using the following procedure:
>
> Input: An undirected graph $G = (V, E)$, where $V$ is the set of nodes and $E \subseteq \{\{u, v\} \mid u, v \in V, u \neq v\}$ is the set of undirected edges.
>
> Output: A directed acyclic graph $\mathrm{DAG} = (V, E')$.
>
> * Assign a unique index to each node: $\mathrm{idx}[v] \gets$ index of $v$ in a fixed ordering of $V$.
> * Initialize the DAG with all nodes $v \in V$.
> * Initialize the edge set: $E' \gets \emptyset$.
> * For each edge $\{(u, v)\} \in E$:
>          If $\mathrm{idx}[u] < \mathrm{idx}[v], E' \gets E' \cup \{(u, v)\}$
>          Otherwise,  $E' \gets E' \cup \{(v, u)\}$.
> * Return the DAG: $\mathrm{DAG} = (V, E')$.
>
>
> The output is a directed acyclic graph $\mathrm{DAG} = (V, E')$, which admits a valid topological ordering and can be used as input to a pLSTM or other DAG-based neural architectures. In our experiments, we use a random fixed ordering for the graph benchmarks. Note that this ordering is non-unique. Future work could address this limitation by using multiple random orderings instead of one.
> The 2D grid admits a special ordering that is self-similar, which we use here.
>
>
> Our manuscript has improved substantially by addressing and incorporating your feedback. If any questions remain, we are happy to engage in further discussion. If you find our responses and additional results useful, we would appreciate it if you would consider changing your score.

---

> ### Comment · Reviewer_hoGY · 2025-08-05
>
> I thank the authors for their detailed response. I particularly appreciate their discussion of my main concerns: potential extensions of pLSTM for other stabilization modes and how their parallelization method relates to previous works. Beyond that I would like to offer a few remarks.
>
>
> - **Association of recurrent states with edges:** I was hoping that the notation could be simplified further by directly working with the line graph representation, but this design choice is ultimately up to the authors.
>
> - **pLSTM vs. Mamba-based models:** Thank you for adding a discussion regarding Mamba and Transformers on graph data. While I’m not very familiar with the literature, I found for example [1] which seems to apply the LRU in a graph setting without randomly chosen paths in the graph.
>
> - **Limited evaluation spatial reasoning:** Thank you for comparing pLSTM to more baselines, particularly “V2M + Mamba2D” on the arrow task; this addresses my concern regarding limited evaluation of spatial reasoning.
>
> - **Limited Evaluation graph data:** The results on Graph data, however, are not particularly insightful since the used baselines are quite outdated. For example, the baselines used in [1] are more recent and also seem to outperform pLSTM on first glance although the experimental setup like model size etc might differ.
>
> Overall I believe that the pLSTM graph mixing framework with its novel stabilization analysis and parallelization technique poses a significant and unique contribution for the field. Although the evaluation is still limited, I believe that the results are promising especially since confounding factors such as backbone and hyperparameters generally favour established architectures over emerging ones. I would, however, recommend to the authors to further shift the focus from introducing a new architecture (which evokes an expectation for sota results on benchmarks) to introducing a novel framework for stabilization/parallelization (where promising results on benchmarks suffice). Since the authors addressed most of my concerns regarding the framework perspective, I will increase the score to 5.
>
> ---
>
> [1] Ding et al. (2024), *“Recurrent Distance Filtering for Graph Representation Learning”*, https://arxiv.org/abs/2312.01538

---

### Note · Authors · 2025-08-11

With pLSTM, we introduce a new way to combine ideas from multi-dimensional RNNs with the efficiency and simplicity of modern linear RNNs. Our main contributions are:

1. __Edge-focused gating mechanisms__ operating on the line graph, enabling a hierarchical parallelization scheme for arbitrary DAGs.
2. __Long-range stability analysis__ leading to two distinct operational modes: directional P-mode and diffusive D-mode. To the best of our knowledge this is the first work to tackle the vanishing/exploding gradient/activation problem in multiple dimensions.
3. __The Arrow Pointing Extrapolation (APE)__ task, a synthetic benchmark for multi-dimensional long-range directional extrapolation.

During the rebuttal, we addressed the main reviewer concerns and requests:

1. __Clarifying novelty vs. Mamba-based extensions__: We clarified that pLSTM generalizes to arbitrary DAGs via a novel parallelization scheme, and offers a more general stability analysis leading to P- and D-modes.
2. __Additional experiments__:

    (a) On ImageNet-1k: With improved initialization, pLSTM performance now closely matches isotropic competitors such as ViL and ViT. In contrast, Mamba-based models employ non-isotropic backbones, which have shown benefits (e.g., in V2M).

    (b) On Graphs: Extended results to Peptides-func and Peptides-struct from the Long Range Graph Benchmark, showing competitive performance. Structural encodings (e.g., LapPE, RWSE) were intentionally omitted to highlight the architecture’s intrinsic inductive bias.

    (c) On APE task: Added results for EfficientNet, ViL, Mamba2D, 2DMamba, V2M, and V2M+Mamba2D, confirming that pLSTM generalizes best to larger extrapolated instances.

3. __Efficiency analysis__: Provided a FLOPs/IO complexity breakdown, showing pLSTM’s linear scaling, compute–memory tradeoffs in parallelization, and training time comparisons for the APE task.

4. __Confirming compatibility with structural encodings__ (e.g., LapPE, RWSE), which can be directly integrated to further boost performance (as typically used in Graph Transformers).

We thank the reviewers for their constructive feedback and the positive joint recommendation for acceptance.

---

### Decision · Program_Chairs · 2025-09-17

**Decision:**

Accept (poster)

**Comment:**

This paper introduces pLSTM, a novel framework that generalizes linear RNNs to directed acyclic graphs (DAGs) via edge-focused gating and a hierarchical parallelization scheme, and provides a stability analysis yielding two operational modes (P-mode and D-mode).

During the discussion, the reviewers raised concerns about how the approach differs from Mamba-based approaches, the limited and somewhat outdated baselines (especially for graph data), and the lack of demonstrated efficiency gains given the motivation of linear RNNs. The authors responded thoroughly with new results (on ImageNet, APE, and peptide graph benchmarks), clarified novelty vs. prior work, corrected notational issues, and provided detailed complexity/runtime analyses. They also emphasized that pLSTM is a general framework rather than an optimized benchmark competitor.

Most reviewers agreed that the framework is conceptually strong and opens new directions, even if benchmark performance is not yet state of the art. Given the conceptual advances I believe asking for state-of-the-art performance is not necessary and recommend accepting the paper.